

# Application of machine learning techniques for warfarin dosage prediction: a case study on the MIMIC-III dataset

Aasim Ayaz Wani[1] and Fatima Abeer[2]

[1] School of Engineering, Cornell University, Ithaca, New York, United States
[2] Jahurul Islam Medical College, University of Dhaka, Bhagalpur, Bangladesh

## ABSTRACT

Warfarin, a commonly prescribed anticoagulant, poses significant dosing challenges due to its narrow therapeutic range and high variability in patient responses. This study applies advanced machine learning techniques to improve the accuracy of international normalized ratio (INR) predictions using the MIMIC-III dataset, addressing the critical issue of missing data. By leveraging dimensionality reduction methods such as principal component analysis (PCA) and t-distributed stochastic neighbor embedding (t-SNE), and advanced imputation techniques including denoising autoencoders (DAE) and generative adversarial networks (GAN), we achieved significant improvements in predictive accuracy. The integration of these methods substantially reduced prediction errors compared to traditional approaches. This research demonstrates the potential of machine learning (ML) models to provide more personalized and precise dosing strategies that reduce the risks of adverse drug events. Our method could integrate into clinical workflows to enhance anticoagulation therapy in cases of missing data, with potential applications in other complex medical treatments.

# INTRODUCTION

## Background

Warfarin, a widely prescribed oral anticoagulant, is essential for preventing blood clots despite the availability of alternatives. In 2019, it accounted for 29.6% of oral anticoagulant use among U.S. Medicare patients (*Troy & Anderson, 2021*). However, managing warfarin is challenging due to its narrow therapeutic range and variability across patients (*Shah, 2020*). Its effectiveness is monitored *via* the international normalized ratio (INR), with most patients needing an INR between 2.0 and 3.0 to avoid bleeding or thromboembolic events (*Ramasamy et al., 2020*). Factors like genetics, diet, and drug interactions add complexity to dosing (*Duarte & Cavallari, 2021*), contributing to warfarin's role in a high rate of emergency hospitalizations for adverse drug events in older adults (*Budnitz et al., 2011*). Machine learning (ML) offers a promising solution by analyzing complex data to

Corresponding author
Aasim Ayaz Wani,
aw579@cornell.edu

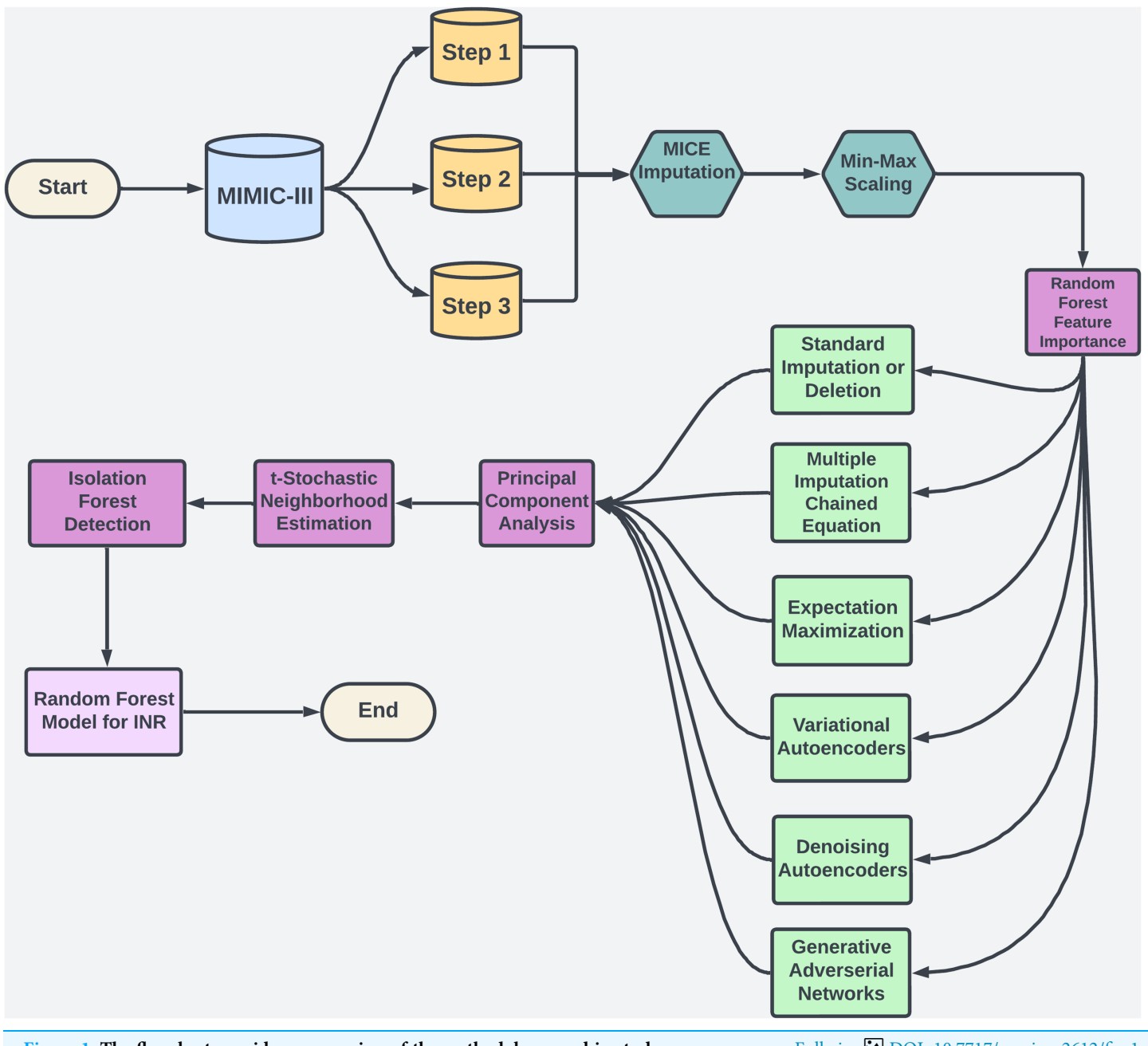

**Figure 1** The flowchart provides an overview of the methodology used in study.

personalize dosing, as illustrated in Fig. 1, potentially improving dosing accuracy and patient safety.

## Objective

Accurately dosing warfarin is critical for preventing adverse drug events, but traditional models often struggle with the variability in patient responses and the complex relationships found in clinical data (*Xue et al., 2024*). In addition, electronic health records

(EHRs) often contain missing data, and this missingness is frequently missing not at random (MNAR), meaning that the absence of data itself may carry clinical significance (*Mitra et al., 2023*). For instance, missing INR measurements during stable periods can skew predictions and lead to biased results (*Yoon, Jordon & Schaar, 2018*). Standard imputation methods, such as mean or median substitution, often fail to address these complexities adequately, resulting in inaccurate predictions (*Beaulieu-Jones et al., 2017*). The goal of this study is to improve the accuracy of warfarin dosing by applying advanced ML techniques that can handle the challenges of missing data in high-dimensional clinical datasets. Focusing on the MIMIC-III database (*Johnson et al., 2016*), this research evaluates the effectiveness of deep learning models such as denoising autoencoders and generative adversarial networks for reconstructing missing data (*Yoon, Jordon & Schaar, 2018*). These models are particularly useful for filling gaps in critical variables like INR, which are essential for precise dosing. By leveraging these advanced techniques, the study aims to produce more reliable INR predictions, enhance warfarin dosing strategies, and ultimately reduce the risk of adverse drug events.

## Challenges of missing data in MIMIC-III dataset

The variability of patient responses and the presence of missing data in clinical datasets like MIMIC-III, present significant challenges for accurately predicting warfarin dosage. Missing data is particularly problematic because it is often MNAR, where the missingness is related to unobserved clinical factors (*Mitra et al., 2023*). For example, missing INR measurements during stable periods could reflect underlying stable conditions, yet they introduce bias when predicting instability (*Yoon, Jordon & Schaar, 2018*). Inconsistent data entry, irregular test-ordering practices, and the severity of a patient's condition further exacerbate these issues. Traditional imputation methods oversimplify variable interactions and do not adequately address the complex, non-linear relationships within clinical data. These methods fail to model interactions between critical factors, such as patient demographics, comorbidities, and medication interactions, which are crucial for accurate warfarin dosing (*Beaulieu-Jones et al., 2017*). The limitations of these methods often result in biased predictions and improper dosage adjustments. Advanced ML models, such as denoising autoencoders (DAEs), variational autoencoders (VAEs), and generative adversarial networks (GANs), offer more effective solutions by capturing the underlying complexity in the data and handling MNAR patterns more appropriately (*Yoon, Jordon & Schaar, 2018*). These models leverage temporal relationships and generate plausible imputations, improving the accuracy of INR predictions and enhancing clinical decision-making.

## Problem statement

Missing data in high-dimensional clinical datasets, such as EHRs, leads to inaccurate INR predictions and improper warfarin dosing (*Bolón-Canedo, Sánchez-Maroño & Alonso-Betanzos, 2016*). Traditional imputation methods fail to capture complex, non-linear relationships, resulting in biased predictions (*Beaulieu-Jones et al., 2017*). This study applies ML techniques to model these patterns, improving the reliability of dosing models & reducing adverse drug events (*Yoon, Jordon & Schaar, 2018*). Using the MIMIC-III

dataset, these proposed imputation techniques enhance warfarin dosage accuracy and improve patient outcomes. Beyond warfarin, this research offers a framework for handling missing data in clinical datasets, benefiting healthcare broadly (*Mitra et al., 2023*). Integrating ML into clinical workflows (see Fig. 1) holds potential for personalizing treatment, reducing hospital readmissions, & improving patient quality.

### Rational for INR prediction

Predicting INR values, rather than directly forecasting warfarin dosage, provides critical advantages for individualized anticoagulation management. INR is a standardized measure of blood clotting with a well-defined therapeutic range (typically 2.0–3.0), which minimizes risks of adverse events such as bleeding or thrombosis. Focusing on INR enables precise, patient-specific dose adjustments that reflect individual variations in response to warfarin, influenced by factors like genetics, diet, and concurrent medications. This approach aligns seamlessly with clinical practice, where INR guides dosing adjustments rather than relying on a fixed dosage—an inherently variable factor across patients. By directly targeting INR, the model reduces complexities involved in dosage prediction, leveraging a universally accepted outcome that clinicians routinely use. Furthermore, INR prediction enhances model interpretability, as it is anchored to validated clinical thresholds that make predictions more actionable and reliable for healthcare providers. Ultimately, INR-based predictions support a safer, patient-centered approach to warfarin therapy, bridging the model's outcomes with real-world anticoagulation protocols.

### Organizational structure

The article is organized as follows: "Literature Review" reviews related work and gaps, establishing the need for advanced ML in warfarin dosing. "Methodology" covers data and preprocessing, highlighting EHR challenges. "Results" details the methodology, focusing on ML-based imputation. "Results" presents results, comparing traditional and advanced techniques. Finally, "Conclusions" discusses the findings, clinical implications, and future research.

## LITERATURE REVIEW

Warfarin, a widely prescribed anticoagulant, requires precise dosing due to its narrow therapeutic index and significant variability in patient responses. Genetic factors, diet, and comorbidities further complicate warfarin dosing, with under-anticoagulation leading to thromboembolic events and over-anticoagulation increasing the risk of bleeding complications. Traditional dosing methods, such as clinical algorithms incorporating pharmacogenetic data and patient characteristics, have demonstrated limitations in capturing the complex, non-linear interactions between variables. Consequently, these methods often struggle to provide individualized dosing recommendations, especially when dealing with high-dimensional clinical data (*Beaulieu-Jones et al., 2017*; *Waljee et al., 2013*). ML techniques, capable of learning from complex datasets and optimizing warfarin dosing at an individual level, offer a promising alternative approach that has shown potential in overcoming these challenges (*Roche-Lima et al., 2020*; *Steiner et al., 2021*).

A critical factor influencing warfarin dosing variability is genetics. Polymorphisms in the genes *CYP2C9* and *VKORC1* have been widely studied for their role in warfarin metabolism and sensitivity. For example, polymorphisms in *CYP2C9*, such as the 2 and 3 alleles, are associated with reduced enzyme activity and subsequently lower warfarin dose requirements (*Beaulieu-Jones et al., 2017*). Similarly, the *VKORC1* gene's -1639GA polymorphism has been linked to lower dose requirements. In addition to genetic factors, age, body surface area, comorbidities, and concomitant medications also play crucial roles in determining the optimal warfarin dose (*Waljee et al., 2013*). ML models, including random forests and support vector machines, have demonstrated superior performance over traditional methods by accommodating non-linear relationships between these variables. For example, studies have shown that ML-based dosing models improve predictive accuracy, particularly in underrepresented populations like Caribbean Hispanics and Latinos, where genetic variability plays a significant role in warfarin metabolism (*Roche-Lima et al., 2020*; *Choi et al., 2023*).

Despite the potential of ML techniques, handling missing data in clinical datasets remains a significant challenge. EHRs often suffer from missing values, which can introduce bias and reduce the performance of predictive models. Traditional imputation methods, such as mean imputation, while computationally simple, fail to capture the intricate dependencies between clinical variables. For instance, *Beaulieu-Jones et al. (2017)* demonstrated that mean imputation resulted in suboptimal predictions, with a mean absolute error (MAE) of 0.586 for missing INR values. In contrast, more sophisticated methods like MICE achieved a lower MAE of 0.332, highlighting the need for advanced imputation techniques that can better model complex data relationships.

The evolution of imputation techniques has significantly improved the handling of missing data in clinical settings, particularly with datasets like MIMIC-III. Early methods such as k-nearest neighbors (KNN) and Multivariate Imputation by Chained Equations (MICE) were widely used but often struggled to model the temporal and cross-variable relationships inherent in healthcare data. For example, *Waljee et al. (2013)* found that MICE outperformed mean imputation for missing INR values, achieving a root mean square error (RMSE) of 0.37 compared to 0.51. However, MICE still struggled with non-linear interactions and time-series data, necessitating the development of more advanced approaches, particularly when paired with ML techniques (*Che et al., 2018*). Methods like those by *Wang et al. (2023)* show the effectiveness of advanced algorithms for high-dimensional sparse data imputation. Although focused on spatiotemporal data in crowd-sensing, their approach highlights principles relevant to managing sparsity and missing data in clinical datasets like MIMIC-III.

Recent advances in ML have introduced more effective methods for imputing missing values in multivariate time-series data. For example, *Qin & Wang (2023)* proposed an end-to-end generative adversarial network (E2GAN) model, which achieved an MAE of 0.268 for imputing missing INR values in the MIMIC-III dataset, significantly outperforming traditional methods such as MICE (MAE 0.332) and even the GRU-D model (MAE 0.280). GAN-based models excel at capturing intricate dependencies within clinical datasets and offer more accurate imputations by leveraging adversarial training frameworks. ML

techniques, capable of learning from complex datasets and optimizing warfarin dosing at an individual level, offer a promising alternative approach that has shown potential in overcoming these challenges (Roche-Lima et al., 2020; Steiner et al., 2021). For instance, Petch et al. (2024) demonstrated the effectiveness of ML models in optimizing warfarin dosing specifically for patients with atrial fibrillation, further highlighting the clinical relevance of these advanced techniques (Petch et al., 2024). Similarly, Che et al. (2018) developed the GRU-D model, a variant of recurrent neural networks (RNNs), which effectively captured long- and short-term dependencies in clinical data. Despite their success, these models are not without limitations. High computational costs and the risk of overfitting, particularly with smaller datasets, remain significant challenges. Moreover, the lack of clinical interpretability in deep learning models, such as GANs, complicates their integration into healthcare workflows (Lan et al., 2020).

The versatility of ML-based imputation methods extends beyond warfarin dosing to other clinical prediction tasks. For instance, Harutyunyan et al. (2019) developed a multi-task deep learning model using MIMIC-III to predict patient mortality, length of stay, and ICD-9 code classification. Similarly, Johnson et al. (2017) applied MIMIC-III data to sepsis onset prediction, demonstrating the dataset's utility in advancing clinical decision support systems. The application of ML techniques in these tasks underscores their potential to revolutionize personalized medicine. However, ensuring the scalability and interpretability of these models remains a challenge, particularly as healthcare datasets grow complex.

In conclusion, while traditional methods have laid the foundation for warfarin dosing, ML techniques offer substantial improvements in handling the complexities of clinical data. Advanced models, such as E2GAN and GRU-D, significantly reduce prediction errors compared to traditional imputation methods, offering promising pathways for personalized medicine. However, challenges related to computational demands, model interpretability, and clinical integration remain, requiring further research and development. As ML models continue to evolve, their impact on healthcare will likely expand, paving the way for more precise, personalized treatment strategies across various medical conditions.

## METHODOLOGY

This section outlines the steps to develop a machine learning model for predicting INR using MIMIC-III data. We begin with data preparation, including merging, scaling, and dimensionality reduction to manage high-dimensional data. Missing data is handled with imputation methods tailored to MCAR, MAR, and MNAR types. Isolation Forest removes outliers, and Random Forest predicts INR. Consistent performance metrics at each stage enhance accuracy and reliability, creating a model suited for clinical application.

### Dataset overview and clinical relevance

This study uses the MIMIC-III v1.4 database, a publicly available resource with de-identified health data from 61,532 ICU admissions (Johnson et al., 2016). To comply with Health Insurance Portability and Accountability Act (HIPAA), personal identifiers like

**Table 1 Overview of MIMIC-III tables used for data merging, table shapes & clinical information.**

| Table name | Table dimensions | Description |
| --- | --- | --- |
| Lab events | (27,854,055, 9) | Contains sequential measurements of laboratory test results. |
| Chart events | (7,642,110, 3) | Records patient demographic information, entries/exits to ICU. |
| Output events | (434,921,714, 14) | Contains records of outputs such as medications, test results & clinical interventions, *e.g.*, timestamps & procedures. |
| Patients | (46,520, 9) | Provides demographic details *e.g.*, age, gender, & ethnicity. |
| Input mv | (3,618,990, 31) | Details about medical procedures and medication administered. |

names, contact details, and exact admission dates were removed, preserving detailed medical records for analysis. MIMIC-III includes 28 interconnected tables with a broad range of clinical variables, such as demographics, vital signs, lab results, procedures, medications, and outcomes (see Table 1). For this study, tables like *Lab Events*, *Patients*, *Output Events*, and *Input Events* were linked by primary and secondary keys (*subject id*, *hadm id*, *icustay id*) to track patient trajectories and support model development and evaluation. One key strength of MIMIC-III is its longitudinal structure, which allows dynamic assessments of patient conditions over extended ICU stays (*Chua et al., 2021*). This temporal depth is essential for ML models analyzing patient trajectories. Furthermore, MIMIC-III's coverage across various ICU settings, like surgical and cardiac units, enhances the generalizability of research findings, making them relevant to real-world clinical environments (*Yang et al., 2023*), thus ensuring broad applicability for improving patient outcomes and treatment protocols.

## Data preprocessing and merging techniques

For this study, several preprocessing techniques were applied to the MIMIC-III dataset to maintain data integrity, manage missing values, and ensure robust feature selection for ML. This process included merging techniques, data scaling, and handling temporal data to enhance interpretability and minimize bias. The MIMIC-III dataset uses *ItemID* as a unique identifier to extract medical measurements, clinical events, and lab results. For example, in the *labevents* table, *ItemID* identifies tests like serum potassium (*ItemID 50971*) or creatinine (*ItemID 50912*), enabling efficient extraction of data from thousands of ICU admissions. Additionally, *ItemID* links test values to patient records and timestamps, which is crucial for analyzing temporal trends in patient health. To ensure data completeness and retention, both outer and inner merging methods were employed at various steps. See Fig. 2 and Table 2 for details. Each merging technique had trade-offs. Outer merging preserved rare cases but increased missing data, requiring advanced imputation like MICE. Inner merging reduced missing data but limited generalizability by excluding rare cases. SQL queries were used to extract patient demographics, vitals, and lab results, with *ItemID* referencing biochemical parameters.

- **Step 1: Inner merging** was used to focus on coagulation and biochemical factors, ensuring that only complete records across key variables were included. Although this

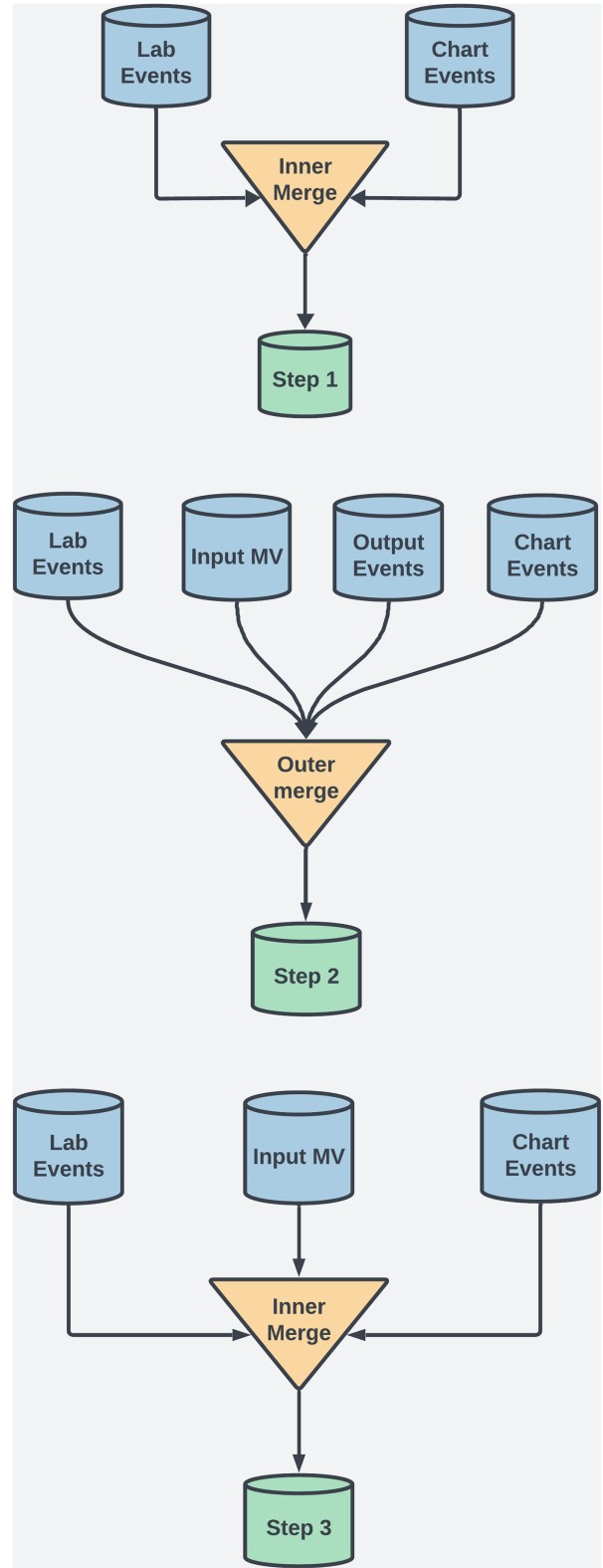

**Figure 2** This diagram illustrates the dataset creation across three steps in the study.

**Table 2  Shape of the merged MIMIC-III dataset across various steps of the study.**

| Step no. | Merging type | Database shape |
|---|---|---|
| Step 1 | Inner merging on coagulation and biochemical data | (47,503, 13) |
| Step 2 | Outer merging on coagulation and biochemical data | (5,000,000, 34) |
| Step 3 | Inner merging on some coagulation and biochemical data | (61,453, 17) |

reduced error for the ML models, it also excluded rare case, a necessary tradeoff to obtain comprehensive feature sets.

- **Step 2: Outer merging** was applied to primary coagulation factors using "Subject ID" and "Chart Time" as keys. This approach retained all patient records, including those with missing data, ensuring that rare cases, which could offer valuable clinical insights, were preserved for analysis.

- **Step 3: Inner merging** was re-applied, combining selected coagulation and biochemical factors. This balanced data completeness with the inclusion of critical clinical markers across a broader patient population.

Before proceeding, we addressed missing values with temporary assignments, prioritizing imputation methods that avoided strict data assumptions for this initial analysis. After exploring alternatives, we selected MICE for its flexibility in generating multiple imputed datasets, which preserved data variability and minimized bias. This method helped maintain the dataset's integrity without overfitting to a specific distribution. While we also considered alternative methods like KNN and standard imputation, they proved less effective in handling the dataset's complexity. Deep learning was not pursued at this step, as their stronger assumptions about data distributions may not suit the inherent variability in clinical data. Min-Max Scaling was applied to normalize clinical variables to a range of 0 to 1, preserving proportional relationships crucial for treatment decisions. We compute the min-max values on the training data only, then apply these parameters to scale the test set independently. This method also minimized distortions from outliers, making it preferable to Z-score normalization. MICE handled missing values, and time-series data was averaged for each patient to improve manageability and interpretation in downstream models, though this reduced temporal dependency.

## Dimensionality reduction methods

Dimensionality reduction is crucial for simplifying high-dimensional datasets by reducing the number of variables while retaining essential information, improving model interpretability, computational efficiency, and visualization. In this study, we applied principal component analysis (PCA) and t-distributed stochastic neighbor embedding (t-SNE) due to their complementary strengths, especially given the high dimensionality and significant missing values in our dataset. PCA transforms correlated variables into uncorrelated principal components, capturing the most variance in the data while

mitigating the effects of missing values by emphasizing the major patterns (*Abdi & Williams, 2010*). This improves model performance and aids visualization by projecting data into lower dimensions (*Hasan & Abdulazeez, 2021*). However, PCA assumes linearity, which can be limiting for non-linear relationships (*Jolliffe & Cadima, 2016*). To address this, we utilized t-SNE, a non-linear method that preserves local relationships between data points, allowing us to visualize complex non-linear structures, often hidden in high-dimensional data, even in the presence of missing values (*van der Maaten & Hinton, 2008*). While t-SNE excels at visualization, it is computationally intensive and sensitive to hyperparameters, making it primarily useful for visual insights rather than predictive modeling (*Kobak et al., 2019*; *McInnes, Healy & Melville, 2018*).

## Missing value imputation methods

The methods for addressing missing data in the MIMIC-III dataset were selected based on assumptions about the nature of the missing data: missing completely at random (MCAR), missing at random (MAR), and missing not at random (MNAR). Each imputation technique was applied with consideration for the likely data mechanism, enhancing the robustness and validity of the imputations.

### *Methods for missing completely at random*

MCAR assumes that missingness is independent of both observed and unobserved data. Deletion methods like listwise deletion (LWD) and pairwise deletion (PWD) are computationally efficient for MCAR data. LWD removes records with missing values, reducing the sample size, while PWD excludes only the missing data in specific analyses, preserving more of the dataset. However, when MCAR does not hold, these methods can introduce bias and reduce model generalizability (*Little & Rubin, 2019*). Single Imputation methods such as mean, median, and mode imputation are simple techniques for MCAR data (*Donders et al., 2006*). In this study, single imputation was applied to continuous (mean/median) and categorical variables (mode) as a baseline. These methods, however, ignore data variability, potentially leading to bias when MCAR assumptions are violated (*Graham, 2009*). Single imputation techniques such as mean, median, and mode single imputation methods, including mean, median, and mode imputation, are simple techniques suited to MCAR data (*Donders et al., 2006*). These approaches replace missing values with the statistical average of observed data. In this study, single imputation was applied as a baseline for continuous (mean or median) and categorical variables (mode).

### *Methods for missing at random*

MAR assumes that missingness depends on observed data but not on unobserved values, allowing for imputation methods that leverage relationships within available data to estimate missing values. Multiple Imputation by Chained Equations is a widely used approach for MAR data, iteratively modeling each variable with missing data as a dependent variable in a regression, using observed variables as predictors (*Van Buuren & Groothuis-Oudshoorn, 2011*). This process generates multiple imputed datasets to account for the uncertainty in missing values, helping to maintain the integrity of the dataset despite missing data points (*Honaker & King, 2010*). In this study, MICE was applied to

**Table 3 DAE model structure and parameters.**

| Parameter type | Descriptions |
|---|---|
| Hyperparameter search | Parameter grid: *batch size* (32, 64, 128), *corruption level* (0.1, 0.3, 0.5), *Learning rate* (0.0001, 0.001), *training epochs* (50, 100, 150). Conduct randomized search with 5-fold cross-validation for 50 iterations. |
| Optimal parameters | *Corruption level* 0.3, *batch size* 64, *learning rate* 0.001, *Training epochs* 100, *optimizer* Adam. |

the MIMIC-III dataset to impute missing vitals and lab results, effectively reducing bias compared to simpler methods, though it proved computationally intensive due to the high dimensionality of the data (*Sterne et al., 2009*). While MICE is effective for handling MAR data with linear relationships, it may be limited in capturing non-linear interactions among variables. To address this, Gaussian mixture models (GMM) were introduced as a complementary approach capable of modeling complex, non-linear patterns within the data. GMMs assume that the data is drawn from a mixture of Gaussians, allowing them to capture heterogeneity within the MAR data and estimate missing values by calculating the most likely values from the Gaussian components (*Bishop & Nasrabadi, 2006*). By combining MICE's linear model capabilities with GMM's strength in handling non-linear relationships, we aimed to enhance the overall accuracy and reliability of imputed values, especially in a high-dimensional clinical dataset like MIMIC-III.

### Methods for missing not at random

MNAR assumes that the probability of missing data is related to the unobserved values themselves. This makes imputing such data more challenging, as the missingness depends on the data that is not available. Advanced ML models, particularly deep learning techniques, are better suited for handling MNAR data because they can model complex non-linear relationships between variables. **Denoising Autoencoders** DAEs are a type of neural network that excels in reconstructing data with missing values by learning robust representations of the data's underlying structure (*Gondara & Wang, 2018*). In the context of MNAR data, DAEs perform well by capturing the relationships between observed variables and imputing missing values based on these learned representations. In this study, DAEs were applied to the MIMIC-III dataset to impute missing data that was likely MNAR—such as missing INR values. DAEs achieved the lowest RMSE in imputing missing values, significantly outperforming traditional methods like MICE and GMM (*Gondara & Wang, 2018*). (See Table 3 for details on model hyper-parameters tuning and model architecture). GANs, particularly generative adversarial imputation networks (GAIN), are powerful tools for imputing MNAR data. GAIN models generate plausible imputations by learning the underlying data distribution and iteratively improving their accuracy through adversarial training (*Yoon, Jordon & Schaar, 2018*). In this study, GAIN was applied to the MIMIC-III dataset to handle MNAR patterns, providing more accurate imputations for complex clinical variables. (See Table 4 for details on model hyper-parameters tuning and model architecture). VAEs are generative models that combine

| Table 4 GAIN model structure and parameters. | |
|---|---|
| **Parameter type** | **Descriptions** |
| Hyperparameter search | Parameter grid: *batch size* (64, 128, 256), *hint rate* (0.7, 0.9), |
| | *Alpha* (10, 100, 1,000), *learning rate* (0.0001, 0.001, 0.01). |
| | Conduct randomized search with 5-fold cross-validation for 50 iterations. |
| Optimal parameters | *Batch size* 128, *hint rate* 0.9, *alpha* 100, *Learning rate*: $10^{-4}$, *Iterations*: $10^4$ |

| Table 5 VAE model structure and parameters. | |
|---|---|
| **Parameter type** | **Descriptions** |
| Hyperparameter search | Parameter grid: *batch size* (64, 100, 128), *learning rate* (0.0001, 0.001), |
| | *Hidden layers* (1, 2, 3), *optimizer* (RMSProp, Adam). |
| | Conduct randomized search with 5-fold cross-validation for 50 iterations. |
| Optimal parameters | *Batch size* 100, *learning rate* 0.001, *hidden layers* 2, *training epochs* 10 |
| | *Xavier initialization* applies to network weights, *optimizer* RMSProp |

neural networks with variational inference to generate latent representations of the data, making them well-suited for imputing MNAR data (*Kingma & Welling, 2013*). VAEs provide a probabilistic framework that captures the uncertainty associated with missing values and generates plausible imputed data by maximizing the evidence lower bound (*Kingma & Welling, 2013*). (See Table 5 for details on model hyper-parameters tuning and model architecture).

## Methods of outlier detection—isolation forest

Isolation forest (IF) was chosen for outlier detection due to its efficiency in handling high-dimensional data. Unlike traditional methods relying on distance or density measures, IF isolates anomalies through recursive partitioning, constructing isolation trees by randomly splitting features (*Liu et al., 2017*). The path length from the root node to the isolated point indicates anomaly likelihood, with shorter paths suggesting higher anomalousness. The algorithm has a computational complexity of O(n log n), where n is the number of samples, making it highly scalable without the need for pairwise distance calculations (*Ding & Fei, 2013*; *Hariri, Kind & Brunner, 2019*). (see Table 6 for more details about hyper-parameter details).

## Prediction model—random forest

Random forest (RF) is an ensemble method for classification and regression that combines multiple decision trees through bagging to reduce overfitting and enhance predictive stability (*Genuer et al., 2020*). Each tree is typically grown to its full depth unless constrained by hyperparameters like max depth, which can limit tree size to prevent overfitting (*Hastie et al., 2009*). Predictions are averaged across trees, improving accuracy. A key benefit of RF is its ability to assess feature importance, which aids in feature selection and provides insights into data relationships (*Theng & Bhoyar, 2024*). By randomly selecting features at each split, RF reduces tree correlation, making it particularly effective

**Table 6 Isolation forest model structure and parameters.**

| Parameter type | Descriptions |
|---|---|
| Hyperparameter search | Parameter grid: *n_trees* (50, 100, 200), *max_samples* ('auto', 0.5, 0.75), *contamination* (0.01, 0.05, 0.1), *max_features* (0.5, 0.75, 1.0). Search using randomized search CV combined with 5-fold cross-validation and 50 iterations |
| Optimal parameters | *n_trees* 100, *max_samples* 'auto', *contamination* 0.1, *max_features* 1.0, *max_depth* None. |

**Table 7 Random forest model structure and parameters.**

| Parameter type | Descriptions |
|---|---|
| Hyperparameter search | Parameter grid: *n_trees* (50, 100, 200), *max_depth* (None, 10, 20), *min_samples_split* (2, 5), *min_samples_leaf* (1, 2). Search using randomized search CV combined with 5-fold cross-validation for 50 iterations. |
| Optimal parameters | *n_trees* 100, *max_depth* None, *min_samples_split* 2, *min_samples_leaf* 1. |

for complex datasets. While its computational complexity scales with the number of trees, samples, and features, RF generally performs well with careful feature selection, especially in high-dimensional data. Overall, RF offers a robust balance of accuracy, interpretability, and performance across varied tasks, (see Table 7 for hyper-parameter details).

## Methodology and performance evaluation

This study's methodology follows a structured approach divided into three progressive steps, each refining the imputation process and improving predictive modeling. The foundation of each step is the initialization of missing data using MICE. Techniques for dimensionality reduction, outlier detection, and predictive modeling are then applied. The performance of each step is evaluated using consistent metrics, ensuring fair comparisons and highlighting the impact of more sophisticated imputation techniques.

- **Data segmentation and imputation techniques: Step 1:** MICE imputation is combined with basic ML models and outer merging strategies. This step retains all patient records, including those with missing data, preserving rare cases but introducing challenges in handling extensive missing data. Step 2: Advances to more sophisticated imputation methods while continuing with MICE. Inner merging is employed to maintain records with complete data across key variables, leading to a more accurate dataset but excluding some rare patient cases. Step 3: MICE imputation is integrated with ML methods like DAE & GAN to address MNAR data. Balanced merging techniques are introduced to preserve critical clinical markers, resulting in a robust dataset for predictive modeling.
- **Dimensionality reduction and outlier detection:** After imputation, PCA simplifies the dataset by capturing major patterns, while t-SNE explores non-linear relationships. Isolation Forest is used for outlier detection, removing extreme values that could distort predictive accuracy and producing a more homogenous dataset for modeling.
- **Predictive modeling and feature importance:** RF is employed for predictive modeling due to its ability to handle non-linear relationships and its robustness against overfitting.

It predicts INR based on key clinical variables and identifies influential predictors through its feature importance function. Step 1: The initial dataset, processed with basic imputation and outer merging, results in relatively lower accuracy due to the high amount of missing data. Step 2: More sophisticated imputation and inner merging improve predictive accuracy. Key predictors, such as coagulation markers, INR, and genetic factors, are better identified. Step 3: The most refined dataset, imputed with DAEs and GANs, reduced *via* PCA and t-SNE, and cleaned of outliers, produces the most accurate INR predictions. Critical predictors like *CYP2C9, VKORC1*, biochemical factors, and INR levels are highly ranked in the feature importance analysis.

- **Performance evaluation and metrics:** Imputation effectiveness and predictive model accuracy are evaluated using RMSE, chosen for its sensitivity to larger errors, which is crucial in clinical contexts like INR prediction. The RF regression model with consistent hyperparameters is applied across all steps to ensure fair comparison, attributing performance differences directly to imputation effectiveness. RMSE, preferred over MAE and MAPE, emphasizes outliers and maintains precision within small target ranges, making it the most suitable metric here.

## RESULTS

### Feature importance analysis

Feature importance played a pivotal role in validating the quality of imputation methods and ensuring the accuracy of the ML models for predicting INR. Given that warfarin dosing is influenced by well-established clinical predictors, such as genetic markers and biochemical factors, the feature importance analysis was used to ensure the imputation techniques preserved these critical relationships. This validation was essential to ensure that we could reduce the numbers of variables that would be used in the models to accurately predict INR levels and support effective clinical decision-making (See Tables 8–11 for details on some of the important variables being considered). Figure 3 illustrates the correlations between clinical variables. This analysis informed the selection of variables for the ML models by identifying highly correlated variables that might introduce multicollinearity.

#### *Methodology*

RF models were utilized to assess feature importance through the Gini index, ranking features by their influence on INR prediction. This analysis helped verify whether the imputation methods—such as MICE & DAE preserved the relationships between key predictors or introduced bias (*Gondara & Wang, 2018*). After imputation, feature importance analysis confirmed that vital predictors remained dominant, ensuring that the models maintained clinical relevance.

- **Step 1**: Initial analysis sought to confirm the prominence of critical factors like Factor II and Factor VII. However, unexpected variables like platelet smear dominated the results, suggesting that the imputation process had introduced significant bias, distorting the natural relationships and compromising the model's clinical reliability (*Luo et al., 2016*).

**Table 8 Key variables in lab events table.**

| Variable | ItemID | Clinical relevance to warfarin dosage |
|---|---|---|
| INR | 51237 | Direct measure of blood clotting and used for adjusting warfarin dosage. |
| Bilirubin | 50885 | Elevated levels may indicate liver dysfunction, requiring lower Warfarin. |
| Oxygen | 50816 | Abnormal Oxygen levels may indicate respiratory or cardiovascular issues. |
| Urea nitrogen | 51006 | Impaired renal function reduces warfarin clearance. |
| WBC | 51301 | Infections/inflammation affect liver function, altering Warfarin dosage needs. |
| Bicarbonate | 50882 | Changes in bicarbonate levels may affect Warfarin efficacy. |
| Sodium | 50983 | Indirectly influences warfarin, especially in patients with heart failure. |
| Potassium | 50971 | Imbalance can impact warfarin action, especially in arrhythmia patients. |
| Creatinine | 50912 | Less warfarin clearance needs less dosage to prevent high anticoagulation. |
| Hemoglobin | 51222 | Helps detect bleeding risks in patients on warfarin. |
| Platelets | 51265 | Low counts increase bleeding risk, making warfarin dosing more critical. |
| Prothrombin time (PT) | 51274 | Prolonged PT indicates a need for dose adjustments to reduce bleeding risk. |
| aPTT | 51275 | Warfarin affects aPTT levels when co-administered with heparin. |

**Table 9 Key variables in patients table and their relevance to warfarin dosage in MIMIC-III.**

| Variables | Clinical relevance to warfarin dosage |
|---|---|
| Age | Older patients require less warfarin doses due to slower metabolism. |
| Gender | Gender affects metabolism. Women need lower Warfarin doses than men. |
| Admission type | Indicates if patient had cardiovascular admission, often treated with warfarin. |
| Admission diagnosis | Atrial fibrillation, or embolism are diagnoses that require warfarin therapy. |
| Marital status | Marital status could be a proxy for social support. |
| Mortality indicator | Indicates patient mortality. |

**Table 10 Key variables in output events table and their relevance to warfarin dosage.**

| Variable name | Description and clinical relevance to warfarin dosage |
|---|---|
| Warfarin dosage | Provides information on dosage quantity, frequency, administration route. |
| Medicine start time | Captures the time Warfarin dosage was initiated. |
| Medicine end time | Monitors when Warfarin therapy was discontinued. |
| Urine output | Low urine output may indicate kidney problems & slow warfarin breakdown. |
| Drainage volume | Drainage from surgical sites may indicate bleeding risk. |
| Blood transfusions | Indicates blood transfusion events, essential for adjusting Warfarin dosing. |
| Vitamin K | Tracks the administration of Vitamin K, which counteracts Warfarin. |

- **Step 2**: With a more comprehensive set of biochemical and coagulation markers, the feature importance distribution improved. However, some inconsistencies remained—key predictors such as those related to warfarin's effects did not retain their expected significance. While the model showed progress, the imputation method required further refinement to fully capture the complex interactions between predictors (*Che et al., 2018*).

**Table 11 Key variables in input MV table and their relevance to warfarin dosage.**

| Variable name | Description and clinical relevance to warfarin dosage |
| --- | --- |
| Procedures | Captures the type and timing of medical procedures performed on the patient. |
| Fluid input vol | Monitors total fluid intake, which can affect fluid balance & hydration status. |
| Medication input vol | Tracks the volume of all medications administered, including anticoagulants. |
| Nutrition | Nutrition *e.g.*, Vitamin K intake, which antagonizes warfarin's effect. |
| Blood products | Records administration of products like platelets, plasma, & red blood cells. |
| IV fluid input | Tracks intravenous fluid administration, which can impact renal function. |

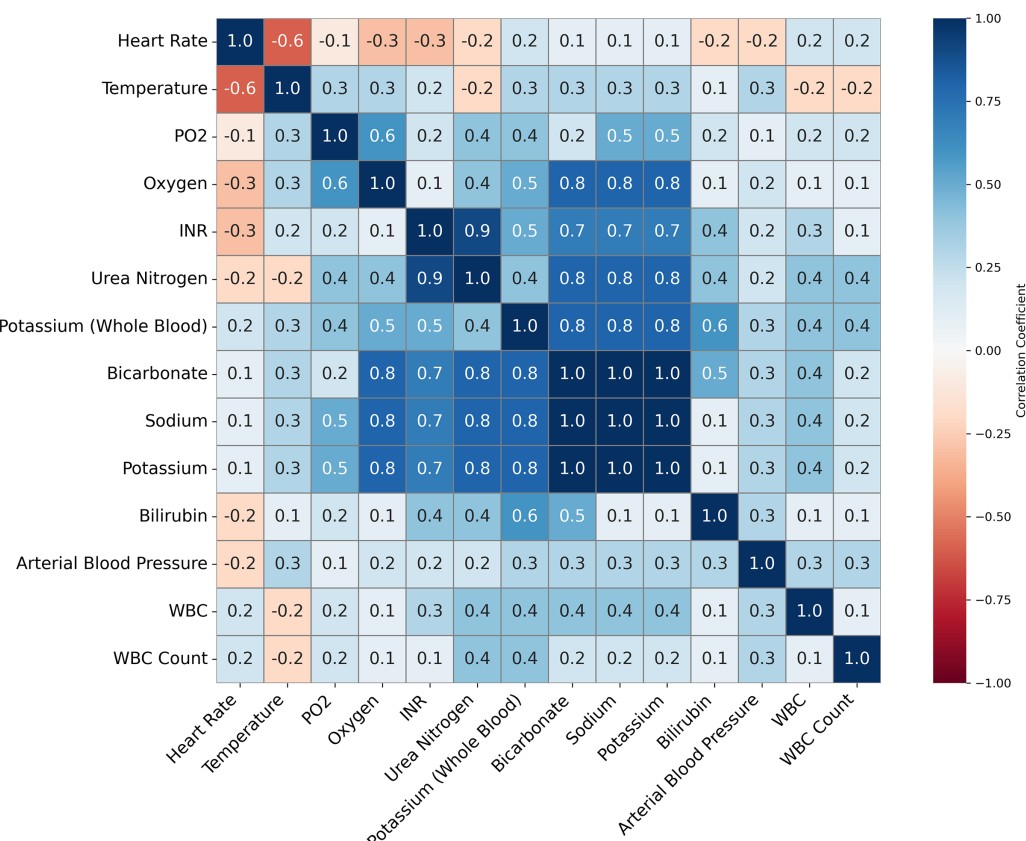

**Figure 3 This figure presents a correlation matrix of selected clinical variables from the MIMIC-III dataset, used to predict INR.** Variables with high correlation, such as Potassium and Bicarbonate, are clustered together. The color scale indicates the strength of the correlations, with blue denoting positive correlations and red denoting negative correlations. This analysis helps identify potential relationships between clinical measures relevant to INR prediction.

- **Step 3**: By refining the imputation approach further, Step 3 aligned well with clinical expectations. Critical factors, such as white blood cell count (WBC), bicarbonate, and gender, were consistently ranked as the most influential predictors (*Xu & Wunsch, 2005*; *Bolón-Canedo, Sánchez-Maroño & Alonso-Betanzos, 2016*). This validation demonstrated that the imputation effectively preserved the necessary relationships, making the predictions more reliable and clinically applicable (*Nazabal et al., 2020*).

### Key insights and clinical implications

The feature importance analysis validated the imputation process and confirmed the accuracy of key clinical predictors across the steps. In Step 1 and 2, the introduction of bias from the imputation process compromised prediction reliability, but Step 3 successfully preserved the integrity of crucial predictors (*Che et al., 2018*). This final step ensured that the models could guide warfarin dosing effectively by maintaining the relationships necessary for accurate INR predictions. Such validation is vital in clinical settings, where warfarin dosing is highly sensitive to specific predictors (*Afkanpour, Hosseinzadeh & Tabesh, 2024*). The insights gained from feature importance analysis underscore the importance of using well-established clinical predictors as benchmarks in ML models for medical applications (*Luo, 2022*). The consistent identification of critical coagulation factors, particularly in Step 3, highlights the model's capability to provide reliable and precise warfarin dosing recommendations. These results serve as a foundation for further refining dosing strategies in clinical practice, ensuring that treatment decisions are personalized and evidence-based (*Duarte & Cavallari, 2021*).

### Clinical application of feature importance

The identification of Factor II, VII, and X as key predictors in our model underscores their importance in warfarin therapy. Variations in these factors significantly impact anticoagulation management, affecting efficacy & safety. Incorporating them into the model allows clinicians to tailor warfarin dosing to individual fluctuations in coagulation, ensuring a more personalized & responsive approach to therapy.

- *Factor II* is key to clot formation as a precursor to thrombin, which converts fibrinogen to fibrin. Warfarin inhibits prothrombin synthesis, prolonging PT and elevating INR. Low prothrombin levels increase the risk of over-anticoagulation and bleeding. Our model identifies Factor II as critical for INR prediction, recommending a 10-15% warfarin reduction to prevent excessive anticoagulation and bleeding.

- *Factor VII* is highly sensitive to warfarin and an early indicator of its effect on coagulation. A decrease in Factor VII rapidly increases PT and INR, making it crucial for immediate dose adjustments. The model's reliance on Factor VII highlights its importance in real-time warfarin management, where fluctuations may signal unstable INR control and the need for frequent monitoring. A sharp decline in Factor VII, even with stable INR, may require a preemptive dose adjustment to prevent thrombotic events.

- *Factor X* is a central component of the common coagulation pathway and directly facilitates the conversion of prothrombin to thrombin. Reductions in Factor X caused by warfarin delay clot formation, leading to prolonged PT and elevated INR. Our model highlights Factor X as a critical factor for fine-tuning warfarin dosing. When a patient's Factor X levels drop more than expected, the model may recommend a slight increase in warfarin to maintain INR within the therapeutic range, thereby reducing thrombotic risk. Conversely, elevated Factor X levels may signal the need to decrease INR to prevent under-anticoagulation.

Consider a patient on long-term warfarin therapy with reduced levels of Factor II and Factor X. The model predicts that such reductions will likely lead to an elevated INR, thereby increasing the risk of excessive anticoagulation and bleeding. In another example, a patient with high Factor VII levels but a subtherapeutic INR may be at risk of inadequate anticoagulation, indicating a need for close monitoring to maintain the INR within the therapeutic range. These cases illustrate how the model's focus on key coagulation factors enables precise, individualized INR predictions, supporting clinical decision-making aimed at improving outcomes and reducing adverse event risks. This predictive approach enhances therapeutic efficacy and minimizes the risks of unstable INR levels, ultimately improving patient safety.

## Dimensionality reduction and outlier detection

To manage the complexity and scale of the MIMIC-III dataset, we implemented a multi-step approach for dimensionality reduction and outlier detection. This process systematically reduced noise, filtered out irrelevant variations, and highlighted critical patterns essential for predictive modeling, with a specific focus on clinical variables like INR. By applying a combination of techniques such as PCA for linear dimensionality reduction, t-SNE for capturing intricate non-linear structures, and IF for detecting and removing extreme outliers, we were able to refine the dataset significantly. Figure 1 illustrates this approach, which facilitated the identification of clusters and outliers crucial for predictive accuracy.

### *Step 1: principal component analysis for initial exploration*

Our analysis began with the application of PCA, a widely utilized technique for linear dimensionality reduction that identifies the directions of maximum variance in high-dimensional data. PCA transforms the dataset into a series of orthogonal principal components, where each component captures a progressively smaller portion of the variance. This method is particularly useful for obtaining a global view of the dataset's structure and simplifying the data while retaining as much variance as possible.

In Step 1 of our analysis, PCA projections revealed distinct clusters in the data, as shown in Fig. 4A. The first two principal components accounted for a significant portion of the variance, indicating that the primary sources of variance in the data were likely driven by consistent patterns in certain clinical measures. For example, patients with similar clinical profiles tended to group into distinct clusters. These clusters suggested that key clinical variables, such as INR, played a critical role in organizing the data along linear axes. This preliminary projection provided a solid foundation for further exploration, offering insight into the global structure of the dataset. The clustering behavior observed in PCA confirmed the potential for predictive modeling based on these linear separations.

However, as the analysis progressed into more complex steps, such as Step 3, the limitations of PCA became apparent. Although PCA effectively captured broad, linear patterns, it became clear that the variance was increasingly distributed across multiple components, and the distinct cluster boundaries observed in Step 1 started to blur

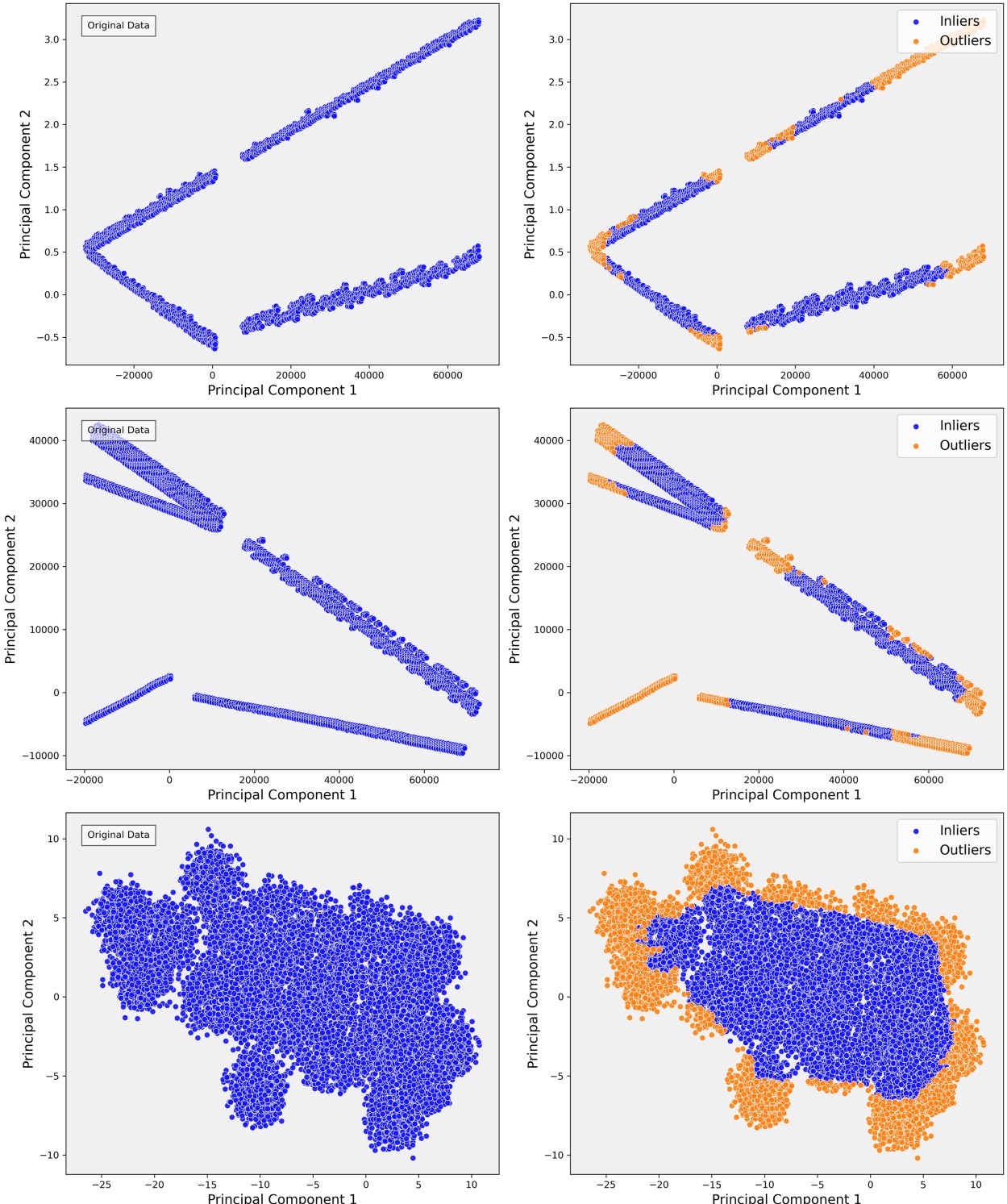

**Figure 4 The scatterplots demonstrate the PCA representation and impact of outlier detection across three steps of the data.** In Step 1, the PCA shows distinct linear bands, reflecting strong directional trends. After applying isolation forest, outliers are primarily removed from the edges, sharpening the core structures. In Step 2, the PCA reveals more dispersed linear patterns, indicating greater complexity. Outlier detection helps refine these structures by removing anomalies that deviate from the main trends, resulting in more cohesive groups. In Step 3, the PCA displays dense, nonlinear clusters, highlighting intricate relationships. Outlier detection effectively separates peripheral points, clarifying the clusters and improving the overall structure of the inlier data.

(Fig. 4C). This indicated the presence of more complex, non-linear relationships in the data that PCA could not capture. For instance, subgroups of patients with rare conditions or outliers were not sufficiently distinguished in the PCA projections. Recognizing these limitations, we transitioned to a more advanced, non-linear dimensionality reduction method t-SNE to capture these subtler relationships.

### Step 2: transition to t-SNE for non-linear structure exploration

Given the complexity of the dataset and the presence of non-linear relationships that could not be fully captured by PCA, we employed t-SNE. t-SNE is particularly effective at preserving local neighborhood relationships in high-dimensional data, making it well-suited for uncovering complex, non-linear structures that are often prevalent in clinical datasets. The t-SNE plots in Fig. 5 illustrate the evolving data structure at different perplexity levels. At low perplexity (5), the data forms a dense cluster, showing minimal differentiation. As perplexity increases (50 to 100), more distinct, maze-like patterns emerge, reflecting clearer separations between data points. At higher perplexities (400 and 500), clusters become more distinct and elongated, capturing both local and global structures, revealing significant subgroups and relationships within the dataset. In Step 3, where the dataset demonstrated greater complexity, t-SNE was applied with a perplexity parameter of 100. This uncovered distinct clusters that were not apparent in the earlier PCA projections (Fig. 6). These clusters appeared to correspond to subgroups of patients with similar clinical profiles, highlighting patterns that were less distinguishable through linear methods like PCA. The finer resolution provided by t-SNE enabled us to identify potential patient subgroups with unique clinical characteristics.

To further investigate these relationships, we increased the perplexity to 400, which revealed even more pronounced separations within the dataset (Fig. 6E). These elongated clusters may represent patients with rare genetic conditions, atypical treatment responses, or other unique clinical factors that were previously obscured within the broader dataset. This level of detail was particularly valuable for identifying potential outliers or patient subsets that warranted closer clinical examination. The ability of t-SNE to reveal such intricate patterns underscored its importance in exploring the non-linear relationships within the data, offering insights that were critical for improving the accuracy of our predictive models.

The visualizations provided by t-SNE played a crucial role in enhancing our understanding of the dataset's non-linear relationships. As perplexity increased, the separation between clusters became more distinct, suggesting higher perplexity values allowed for the identification of more complex patterns within the dataset. This transition from PCA to t-SNE marked a pivotal step in our analysis, enabling us to move beyond the linear structures captured by PCA & delve deeper into the non-linear structures often inherent in clinical data. Overall, application of t-SNE in Step 3 allowed us to explore previously hidden patterns, which were instrumental in identifying patient subgroups and improving outlier detection.

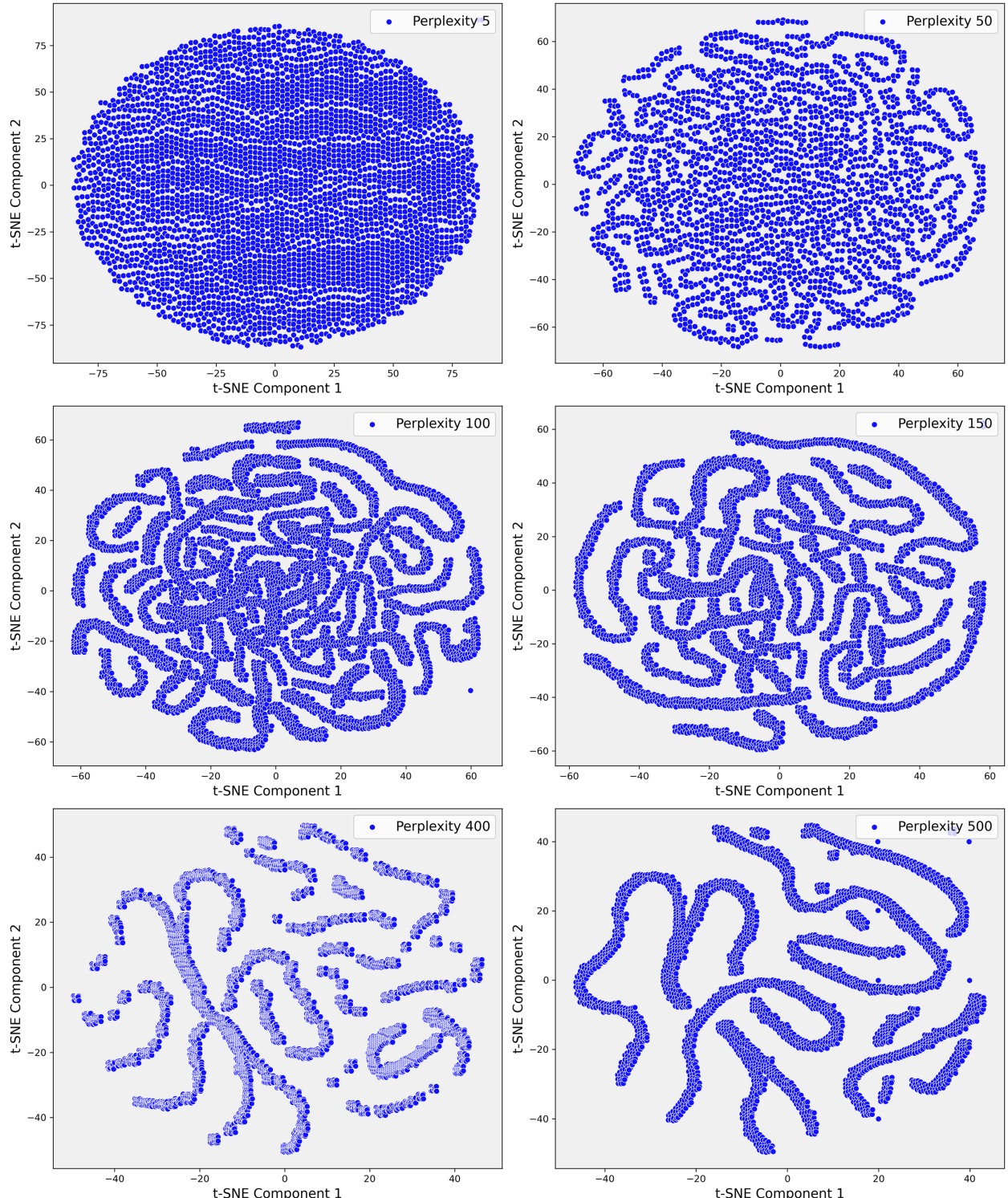

**Figure 5** These scatter plots present the t-SNE representation of Step 1 data across various perplexity values, showcasing the data's underlying structure: (A) Perplexity 5: The data forms a dense, circular cluster with no clear structure. (B) Perplexity 50: Complex, maze-like patterns emerge, reflecting more intricate relationships within the data. (C) Perplexity 100: The patterns become slightly tighter and more defined, resembling those seen at perplexity 50. (D) Perplexity 150: The data continues to display intricate, maze-like structures with varying pattern density. (E) Perplexity 400: More distinct clusters and elongated structures begin to emerge, indicating clearer data groupings. (F) Perplexity 500: The data points form increasingly distinct, elongated structures, highlighting significant patterns and separations within the dataset.

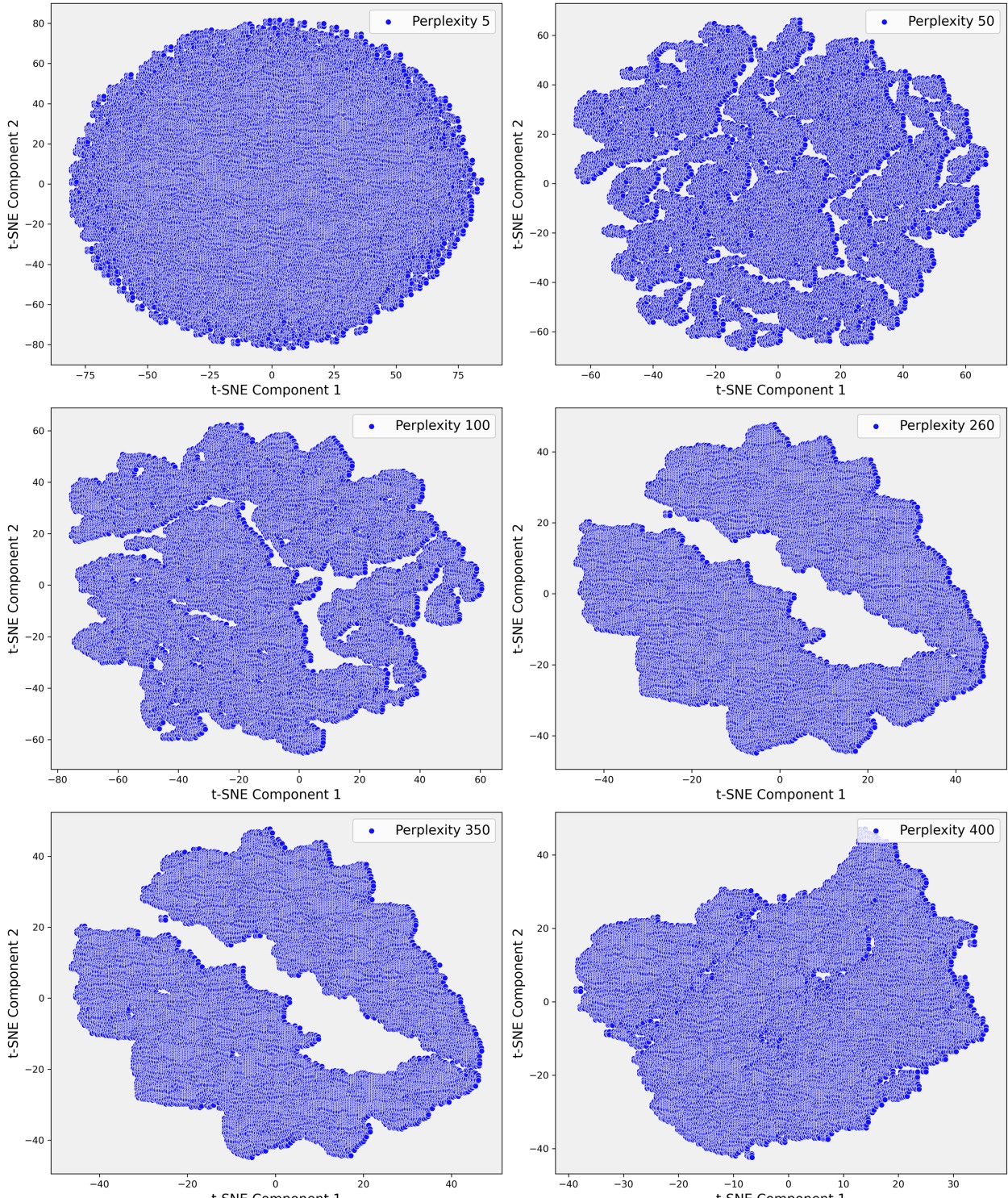

**Figure 6** These scatter plots illustrate the t-SNE representation of Step 3 data across different perplexity values, revealing the evolving structure of the dataset: (A) Perplexity 5: Data points are densely packed in a circular formation with minimal visible patterns. (B) Perplexity 50: Complex, interconnected patterns emerge, indicating more intricate relationships within the data. (C) Perplexity 100: The data shows clearer and more distinct patterns, suggesting emerging structures. (D) Perplexity 260: Elongated, well-separated clusters appear, reflecting deeper divisions within the data. (E) Perplexity 350: The clusters become more defined and distinct. (F) Perplexity 400: Highly separated clusters emerge, highlighting significant underlying patterns and clear separations in the dataset.

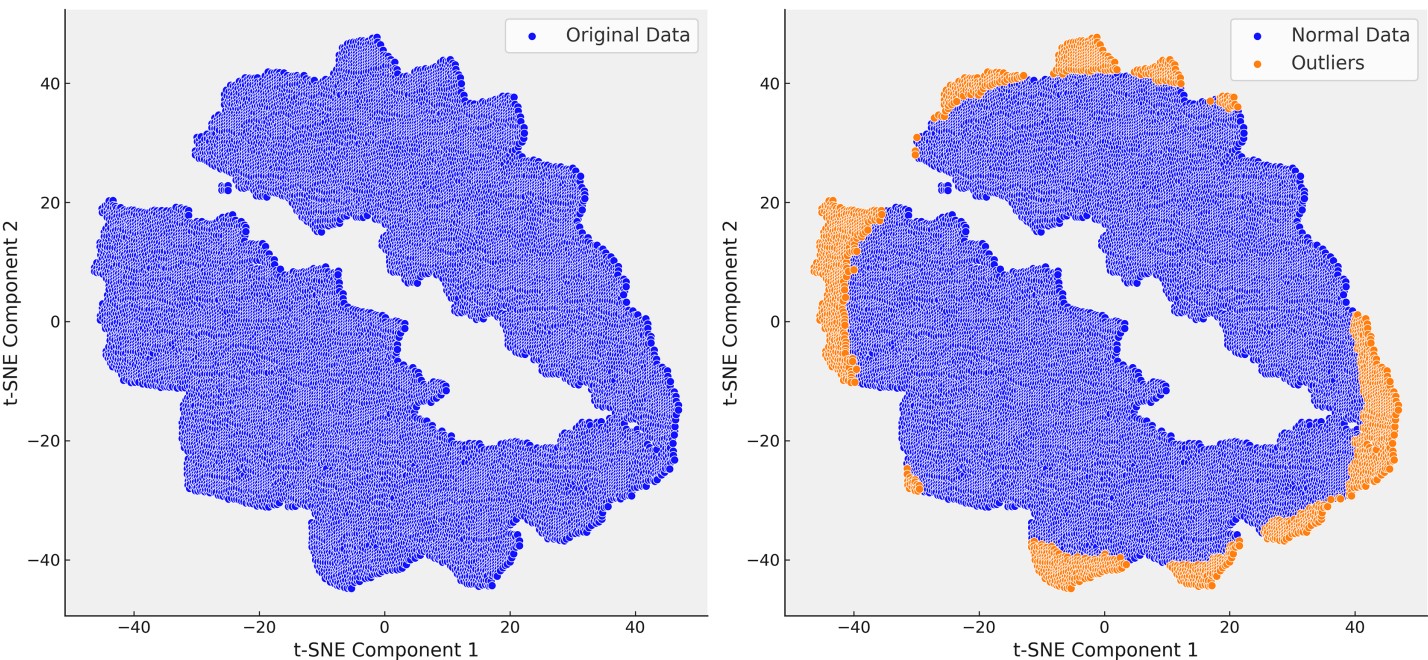

**Figure 7** **In the right t-SNE plot, outliers predominantly cluster along the edges of the main data groups, forming continuous bands around the core clusters.** This suggests systematic deviations from the central patterns, likely representing rare cases, boundary conditions, or data anomalies. The elongated structure of some outliers indicates they share certain features that distinguish them from the main data but are not enough to form separate clusters.               

### Step 3: outlier detection with isolation forest

Following the identification of the dataset's finer structure using t-SNE, we proceeded to the critical step of outlier detection using isolation forest (IF). In clinical datasets like MIMIC-III, outliers often represent patients with rare conditions or errors in data collection, and accurately detecting and handling these outliers is essential for improving the quality of predictive models. Isolation forest, a tree-based anomaly detection algorithm, isolates anomalies by recursively partitioning the data. Unlike traditional distance-based methods, isolation forest is well-suited for high-dimensional and heterogeneous data, such as MIMIC-III, due to its scalability and ability to handle complex data distributions without relying on specific distance metrics. Initially, isolation forest was applied across all steps of the analysis. However, after confirming through t-SNE that Step 3 provided the most reliable separation of patient subgroups, we focused our outlier detection efforts on this step. The clusters revealed by t-SNE served as a natural guide for detecting outliers, allowing Isolation Forest to identify and remove extreme values that distorted the dataset. For example, extreme INR values, such as a mean of 54, were detected and removed, which significantly improved the clarity and homogeneity of the dataset (Fig. 7).

The elimination of extreme cases reduced noise in the data and improved the accuracy of our predictive models. For instance, removing extreme INR values helped refine our warfarin dosing predictions, leading to a more reliable model that better reflected real-world clinical scenarios. The successful application of the Isolation Forest algorithm,

guided by t-SNE, is highlighted in Fig. 7, where the reduction in outliers is clearly visible. This refinement process enhanced the dataset's suitability for subsequent analysis and improved the overall performance of our predictive models.

In conclusion, the combination of PCA, t-SNE, & IF provided a systematic and effective approach to refining the MIMIC-III dataset. PCA captured broad variance and general patterns, while t-SNE revealed non-linear relationships & organized the data into meaningful clusters. Finally, IF identified & removed outliers, enhancing dataset quality & improving model accuracy. This integrated methodology optimized the data for accurate predictions, particularly by improving INR imputation and better modeling patient subgroups, ultimately contributing to clinically relevant & actionable insights.

## Inference drawn from missing value imputation methods

- **Deletion methods**: Deletion methods handle missing data by removing records with incomplete information. While simplifying the dataset, this introduces significant drawbacks in high-dimensional clinical datasets like MIMIC-III, where missing data in critical variables (*e.g.*, lab results and vital signs) is common. Deletion leads to selection bias, overrepresenting patients with fewer complications and excluding those with irregular monitoring, often more severely ill or with rare conditions (*Waljee et al., 2013*; *Beaulieu-Jones et al., 2017*). This bias skews the dataset toward typical cases, reducing variability needed for accurate predictions and oversimplifying clinical responses like INR outcomes (*Sohrabi & Tajik, 2017*). Consequently, deletion methods can result in inaccurate warfarin dosing recommendations, increasing the risk of complications such as bleeding or thrombosis in atypical patients (*Emmanuel et al., 2021*). Additionally, deletion reduces dataset size, weakening statistical power and limiting the detection of significant relationships between patient characteristics and INR outcomes, particularly in critical subgroups (*Murray, 2018*). For instance, in MIMIC-III, 80% of prothrombin time (PT) results and 70% of Glasgow Coma Scale (GCS) scores are missing, skewing predictions toward stable cases. Similarly, 55% of Heparin administration data is missing, directly affecting INR management (*Yoon, Jordon & Schaar, 2018*).

- **Single imputation methods**: Single imputation techniques (mean, median, or mode imputation) applied to the MIMIC-III dataset also fall short in handling missing data, resulting in biased and less reliable INR predictions (*Beaulieu-Jones et al., 2017*; *Waljee et al., 2013*). These methods oversimplify by replacing missing values with statistical averages, distorting natural variability and reducing patient heterogeneity (*Rubin, 2004*). Patients with more complete data often have severe conditions due to intensive monitoring, biasing imputation towards these cases, while patients with milder conditions and sparser data are underrepresented (*Madley-Dowd et al., 2019*). Mean imputation centralizes predictions around the average, overlooking individual differences and leading to inaccurate dosing recommendations (*Beaulieu-Jones et al., 2017*). Median imputation, though more robust to outliers, still oversimplifies patient-specific responses. Mode imputation is unsuitable for continuous measures like INR, further obscuring critical differences in patient responses and increasing the risk of

adverse outcomes (*Waljee et al., 2013*). These methods fail to capture the complexity of clinical data, highlighting the need for more sophisticated imputation approaches for accurate INR predictions (*Nazabal et al., 2020*).

- *Multiple Imputation by Chained Equations (MICE)*: was applied to the MIMIC-III dataset for imputing missing data using an iterative process, where each variable is predicted based on observed values from other variables. MICE worked effectively for common conditions like sepsis and heart failure with MCAR or MAR. However, it struggled with MNAR, especially in sicker patients, leading to biases that compromised the accuracy of INR predictions—critical for warfarin dosing decisions. MICE's assumption that missingness could be explained by other observed variables often faltered in complex cases involving systematic missingness. MICE also encountered difficulties with high-dimensional clinical data, where the relationships between variables are often non-linear. This limitation affected the precision of INR predictions, a metric influenced by multiple interacting factors. Moreover, MICE was not designed to handle temporal dependencies, treating variables as static, which led to inaccuracies in time-sensitive predictions like fluctuating INR levels that are crucial for adjusting warfarin dosages. While MICE outperformed simpler methods like mean/mode imputation, it fell short compared to advanced techniques like DAEs and GANs, which better captured non-linearities and temporal dynamics, resulting in 15-25% lower RMSE and MAE for INR predictions. MICE's reliance on predictor equations also increased the risk of overfitting or mis-specification, particularly in rare or atypical clinical cases. Thorough validation was essential to ensure MICE's imputations did not compromise INR prediction accuracy, especially in high-risk groups where precise warfarin dosing is critical.

- *k-nearest neighbors (KNN)*: imputation was applied to the MIMIC-III dataset, using nearby data points based on clinical features like vital signs and lab values. While effective for common conditions, KNN introduced bias when handling rare conditions or atypical cases by favoring frequent patterns, leading to less accurate imputations. This issue was particularly significant for INR predictions in complex cases where clinical data deviated from norms. Additionally, KNN failed to account for temporal trends, which are essential for accurate INR predictions and warfarin dosing adjustments. Ignoring time-dependent data disrupted patient trajectories, causing inaccuracies in modeling time-sensitive outcomes. Comparative analyses showed KNN performed similarly to simpler methods like mean/mode imputation but was consistently outperformed by deep learning models like DAEs and GANs. These models captured non-linear and time-dependent relationships better, reducing RMSE and MAE by 10–30%. Moreover, KNN was vulnerable to bias in MNAR, disproportionately affecting sicker patients with more incomplete data, skewing INR predictions and increasing the risk of complications like bleeding or thrombosis. Careful validation was needed to ensure imputed values did not compromise clinical accuracy, especially for rare conditions or time-sensitive situations (*Yoon, Jordon & Schaar, 2018*).

- *Gaussian mixture model (GMM)*: GMM imputation was applied to the MIMIC-III dataset to address the complex patterns of missing data by modeling them as a mixture of multivariate normal distributions. While GMM effectively handled non-linear relationships in the data, its performance was mixed when faced with MIMIC-III's unique challenges. For common clinical conditions with well-represented distributions, such as heart failure or pneumonia, GMM produced reasonable imputations. However, for rare conditions or patients with atypical trajectories, GMM often struggled to accurately estimate missing values, as its mixture components could not adequately capture the sparsity and variability inherent in these cases. The absence of temporal modeling in GMM proved a critical limitation in MIMIC-III, where patient trajectories and clinical trends over time (*e.g.*, changes in INR) are essential for accurate predictions. GMM imputation, by treating data points as static, often failed to reflect the time-sensitive nature of clinical variables. This resulted in disruptions to the continuity of patient data, particularly for outcomes like warfarin dosage adjustments that rely on tracking fluctuations in INR over time. Comparative evaluations showed that while GMM outperformed basic imputation techniques such as mean/mode imputation in capturing static relationships, it was consistently outperformed by advanced models like DAE and GANs, which accounted for both non-linear and temporal dependencies. These models achieved a 15-30% reduction in RMSE and MAE compared to GMM, particularly for predicting INR—a key factor in optimizing warfarin dosing in critical care settings. GMM also exhibited biases in cases of non-random missing data (MNAR), a common issue in MIMIC-III, where sicker patients or those with more severe conditions tend to have more incomplete data. GMM's reliance on well-represented distribution patterns resulted in biased imputations, which skewed predictions for these high-risk subgroups. Rigorous validation was necessary to ensure that GMM's imputations did not compromise clinical accuracy, particularly in patients requiring precise monitoring and individualized treatment strategies.

- *Denoising autoencoders (DAEs)*: DAEs were applied to the MIMIC-III dataset to impute missing data by learning latent representations and reconstructing complete data from corrupted inputs. They were particularly effective at capturing non-linear relationships in high-dimensional clinical data, such as vitals, lab values, and patient history, which significantly improved the accuracy of INR predictions compared to simpler methods (*Vincent et al., 2008*; *Gondara & Wang, 2018*). By handling noise and maintaining the integrity of data distributions, DAEs contributed to more reliable INR imputation, which directly impacted clinical decisions, such as warfarin dosing (*Nazabal et al., 2020*). However, in cases of extreme missingness or outliers, DAEs sometimes produced unrealistic imputations, especially when latent features failed to generalize to rare clinical events (*Hastie et al., 2009*). Careful tuning of the model was necessary to mitigate these risks. DAEs also effectively reduced bias in MNAR, a common issue in sicker patients, improving INR predictions in these challenging cases (*Rubin, 2004*). Despite their strong performance, DAEs struggled with temporal dependencies, limiting their ability to accurately predict time-dependent INR fluctuations, which are critical for

guiding warfarin adjustments. Optimizing the model and validating the imputations were essential to ensure accurate INR predictions, particularly in patients with rare conditions.

- *Generative adversarial imputation networks (GAIN)*: a GAN-based model, was applied to the MIMIC-III dataset to impute missing data by leveraging an adversarial framework that captured complex, non-linear relationships in high-dimensional clinical data (*Yoon, Jordon & Schaar, 2018*). The generator modeled missing values while the discriminator distinguished real from imputed data, allowing GAIN to produce realistic imputations that closely matched the original data distributions. This was especially valuable for MNAR data, common in sicker patients, where traditional methods like k-NN and MICE faltered (*Rubin, 2004*; *Sterne et al., 2009*). GAIN excelled in capturing patterns in vitals, lab values, and patient history, contributing to more accurate INR predictions and better-informed warfarin dosing decisions. However, the adversarial nature of GAIN occasionally led to extreme or unrealistic values, particularly when the generator and discriminator dynamics were not well-balanced during training (*Goodfellow et al., 2020*). These distortions were more pronounced in rare clinical cases or when imputing data with extreme missingness patterns, leading to overfitting or the generation of anomalous values that deviated from the expected distribution (*Nazabal et al., 2020*). Careful hyperparameter tuning and validation were essential to mitigate these effects and ensure the reliability of the imputations (*Gondara & Wang, 2018*). Temporal dependencies also presented challenges for GAIN, as it excelled in cross-sectional imputations but was less effective with time-dependent variables such as INR trends, which are critical for guiding warfarin adjustments. Despite these limitations, GAIN consistently outperformed simpler methods, achieving 20-35% reductions in RMSE and MAE (*Yoon, Jordon & Schaar, 2018*). Proper training and validation were crucial to maintain stability, avoid extreme imputations, and ensure accurate clinical predictions, particularly for complex or rare patient profiles (*Hastie et al., 2009*).

- *Variational autoencoders (VAEs)*: We applied VAEs to the MIMIC-III dataset to impute missing data by learning latent representations and using probabilistic models for reconstruction. VAEs excelled in capturing non-linear relationships in clinical data, but the probabilistic nature of the model introduced bias, particularly in cases of non-random missingness from critically ill patients (*Kingma & Welling, 2013*). This bias directly impacted the accuracy of INR predictions, a crucial factor in determining precise warfarin dosing (*Rubin, 2004*). Misjudged INR predictions led to improper dosing, increasing the risk of complications such as bleeding or thrombosis (*Yoon, Jordon & Schaar, 2018*). We also observed that VAEs struggled with temporal dependencies since they were designed for cross-sectional imputation, limiting their ability to model time-dependent trends like fluctuating INR levels (*Murray, 2018*). This reduced the model's effectiveness in predicting INR trends over time, which are essential for making informed adjustments to warfarin dosages based on a patient's evolving clinical condition. Despite these challenges, VAEs demonstrated a strong ability to handle high-dimensional data, though they required significant hyperparameter tuning to optimize

**Table 12 Comparison of root mean square error for Step 1 and Step 3 merging.**

| Method | Step 1 | Step 3 |
| --- | --- | --- |
| Mean, median, mode | 1.2403, 1.2395, 1.24 | – |
| Multiple imputation chained equations | 0.5659 | 0.5659 |
| Expectation maximization | 0.6454 | 0.6454 |
| MissForest | 0.4992 | 0.5837 |
| Generative adversarial networks | 0.007 | 0.09139 |
| Denoising autoencoders | 0.004 | 0.009278 |
| Variational autoencoders | 0.002 | – |

layers and latent dimensions. Validation was essential to ensure that the imputed values were clinically accurate and did not compromise INR predictions (*Donders et al., 2006*). With proper tuning and validation, we improved the accuracy of INR predictions, minimizing the risk of adverse outcomes related to improper warfarin dosing in critical care settings. However, occasional spikes or drops in INR predictions were observed due to the bias and difficulty with non-random missingness, affecting the smoothness of the prediction patterns (*Nazabal et al., 2020*).

Table 12 presents a comparative analysis of RMSE values across the three steps of data imputation and INR prediction. The performance of the models improved significantly as more advanced imputation techniques were applied. Step 1, which utilized simpler methods like mean imputation and KNN, exhibited the highest RMSE values, indicating poor predictive accuracy due to the substantial amount of missing data and the limitations of these basic approaches. In contrast, Step 3, which employed deep learning-based techniques such as DAE and GANs, showed a dramatic reduction in RMSE, reflecting superior handling of MNAR data and more accurate INR predictions. This progression highlights the importance of using advanced imputation techniques to reduce errors in warfarin dosage recommendations from improved INR predictions. The results underscore the necessity of sophisticated models to handle the inherent complexity and missing data patterns in high-dimensional clinical datasets like MIMIC-III.

## DISCUSSION

### Rationale for excluding step 1 and step 2 data

The decision to exclude Step 1 and Step 2 data from the warfarin dosing analysis was driven by both statistical and clinical considerations.

- The mean INR values in both step were alarmingly outside the recommended therapeutic range, suggesting significant issues with data integrity. In Step 1, the mean INR reached 54, while in Step 2, it was 37—dramatically exceeding the standard therapeutic range of 2.0 to 3.0 (or 2.5 to 3.5 for high-risk patients). Such extreme values indicate the presence of substantial outliers or data inconsistencies. Including these outliers in the analysis would introduce significant bias, distort clinical insights, and

compromise the overall reliability of the findings. Consequently, relying on these skewed values for clinical decisions would create unacceptable levels of uncertainty, jeopardizing the accuracy of the model and its ability to produce meaningful and reliable predictions.

- **Dimensionality reduction**: techniques like PCA and t-SNE were applied to differentiate patterns across Steps. Step 1 showed clear clusters with strong linear relationships, while Step 2 presented more dispersion and variability, and Step 3 demonstrated more uniform patterns with reduced outlier influence. This progression highlighted how the underlying data patterns evolved across steps, with Step 3 providing the most stable foundation for model training and prediction.
- The increasing noise and variability observed across steps significantly impacted the analysis. Step 1 data was relatively clean with well-defined clusters, but by Step 2, noise led to greater dispersion and more complex nonlinear relationships, complicating reliable predictions. Despite preprocessing, Step 2's elevated complexity presented challenges for critical tasks like warfarin dosing, while Step 3's reduced noise and improved patterns allowed for more reliable analysis.
- A crucial finding emerged from the feature importance analysis. Key coagulation parameters, such as Factors II, VII, and X—pharmacokinetically critical for determining warfarin dosage—were ranked unexpectedly low in steps 1 and 2. This discrepancy from established clinical knowledge suggested that these steps failed to properly represent the factors driving warfarin metabolism. However, in Step 3, the ranking of these critical features aligned more closely with clinical expectations, indicating that Step 3 better captured the relevant dynamics of warfarin dosing.

Finally, Step 3 data, with a mean INR of 1.5, was much closer to the therapeutic range, making it more clinically relevant and aligned with typical treatment scenarios. Outlier removal and other preprocessing steps enhanced the homogeneity of this step, reducing noise and improving the dataset's suitability for predictive modeling. By focusing on Step 3, we ensured that our analysis was grounded in a dataset that better reflects real-world patient conditions, thereby increasing the reliability and applicability of our warfarin dosing recommendations. In summary, excluding Step 1 and Step 2 data was a necessary step to preserve the integrity of the analysis. Step 3 provided a more robust and clinically relevant dataset, leading to stronger predictive models and more reliable dosing recommendations.

## Interpretability of machine learning models

ML models have improved prediction accuracy and missing data handling in clinical datasets, but their "black-box" nature hinders clinical adoption, as clinicians need to understand how inputs influence outputs for safe and effective treatment. To address this, future work should prioritize interpretability techniques like SHapley Additive exPlanations (SHAP) and Local Interpretable Model-Agnostic Explanations (LIME), which can clarify the impact of features, such as coagulation factors, on INR predictions and make model outputs more actionable for clinicians (*Zafar & Khan, 2021*). For instance, SHAP values could reveal how fluctuations in Factor II affect recommendations,

aiding clinicians in understanding the basis of INR predictions. Balancing accuracy with interpretability is crucial; while advanced models offer strong performance, simpler models like decision trees may be preferable where transparency is essential. A hybrid approach, allowing clinicians to choose between interpretability and performance based on the clinical context, could improve trust and accuracy in ML-driven treatment recommendations.

## Scalability and clinical applications

The scalability of ML models is essential for clinical impact, as their integration into routine workflows requires adaptability across diverse healthcare environments with varying infrastructure and patient populations. While the models in this study show promise for improving warfarin dosing predictions, future work should validate them in large-scale, real-time settings to ensure they can handle high patient data volumes efficiently. Distributed computing, cloud-based architectures, and model compression could reduce computational demands, enabling hospitals with limited resources to deploy these models without extensive on-premise hardware. Real-time frameworks that adjust dynamically to new data will help maintain prediction accuracy, and integrating models into EHR systems could streamline their clinical application. Developing user-friendly interfaces that deliver interpretable, actionable outputs will further support precise, data-driven decisions in time-sensitive clinical environments.

## Limitations and future directions

While this study highlights the potential of advanced ML models for handling missing data in clinical datasets, several limitations persist. One key issue is model interpretability, as GANs and DAEs can act as "black boxes," potentially reducing clinician trust. Future research should investigate explainability methods like SHAP and LIME to enhance clinical transparency (*Zafar & Khan, 2021*). Class imbalance is another challenge, often skewing predictions toward prevalent conditions; techniques like SMOTE and cost-sensitive learning could address this *Afrose et al. (2022)*. Additionally, reliance on the MIMIC-III dataset limits generalizability, warranting validation across multicenter datasets to ensure robustness in diverse clinical environments. Computational complexity also poses a barrier, as resource-intensive models like GANs can be challenging in low-resource settings; optimization techniques, such as pruning and quantization, should be explored. Moreover, future studies could use patients with complete data to rigorously evaluate imputation accuracy by systematically introducing missing values to establish a ground truth. Developing hybrid imputation methods incorporating domain-specific knowledge could also improve accuracy and clinical relevance. Addressing these limitations will make ML models more viable for clinical use, promoting effective integration into healthcare settings to enhance patient outcomes and advance personalized medicine.

## CONCLUSIONS

This study highlights the significant potential of ML techniques, in enhancing INR predictions by effectively managing high-dimensional clinical data with missing values. These models demonstrated improved accuracy in INR prediction, thereby enabling more individualized and precise dosing strategies. By preserving critical patient-specific interactions, the study addresses one of the most challenging aspects of anticoagulation therapy—variability in patient response—laying the groundwork for more personalized medical treatments. However, several challenges remain for the practical implementation of these models in clinical settings. Interpretability, a key factor for clinical adoption, is a notable hurdle with complex ML models such as DAEs and GANs. Future work must prioritize the integration of interpretability techniques like SHAP and LIME, ensuring that clinicians can understand and trust the outputs of these systems. Furthermore, tackling class imbalance in the data and validating the models across diverse, multicenter datasets are essential steps to ensure their robustness and applicability to various patient populations and clinical environments. The scalability of these models also needs further investigation. While our study shows promising results, real-world clinical implementation will require these models to adapt to diverse healthcare settings with varying resource constraints. Future work should explore optimization techniques, including model compression and real-time data integration, to improve computational efficiency without sacrificing accuracy. Additionally, integrating these models into existing EHR systems can streamline their deployment in clinical workflows, supporting healthcare providers in making data-driven, evidence-based decisions. In conclusion, the continued exploration and refinement of ML models in healthcare hold immense potential for advancing patient care. By leveraging advanced imputation techniques, adverse drug event risks can be minimized, significantly improving the quality of life for patients undergoing long-term anticoagulation therapy. Future research should focus on making these models more interpretable, scalable, and generalizable, unlocking new possibilities for personalized medicine across diverse medical conditions.

### Funding
The authors received no funding for this work.

### Competing Interests
The authors declare that they have no competing interests.

### Author Contributions
- Aasim Ayaz Wani conceived and designed the experiments, performed the experiments, analyzed the data, performed the computation work, prepared figures and/or tables, authored or reviewed drafts of the article, and approved the final draft.
- Fatima Abeer analyzed the data, authored or reviewed drafts of the article, and approved the final draft.

## Data Availability

The data is available at the MIMIC-III Clinical Database: https://physionet.org/content/mimiciii/1.4.

## Supplemental Information

Supplemental information for this article can be found online at http://dx.doi.org/10.7717/peerj-cs.2612#supplemental-information.

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
