# Peer review of "Application of machine learning techniques for warfarin dosage prediction: a case study on the MIMIC-III dataset"

_PeerJ Computer Science, doi:10.7717/peerj-cs.2612_

## Round 0.1 · original submission · Major Revisions

Representation of this work is not good. The authors should improve the the result selection.

Reviewer 1 ·

Basic reporting

The paper titled "Machine Learning Solutions for missing data in medical research: Warfarin dosing insights" has been reviewed comprehensively. The paper is poorly written and needs a lot of improvement, Some of the comments are given below:
1. The abstract and conclusion sections need to add a quantitative improvement to your research work.
2. The introduction section needs to add a problem statement and research objective for the proposed work.
3. The figures are of very poor quality; it need to be visible and clearly developed.
4. The literature lacks of latest related work, i.e.,
a. Petch, J., Nelson, W., Wu, M., Ghassemi, M., Benz, A., Fatemi, M., ... & Connolly, S. J. (2024). Optimizing warfarin dosing for patients with atrial fibrillation using machine learning. Scientific Reports, 14(1), 4516.

5. The research work lacks innovation; what is your contribution that has not been addressed before?
6. The results are very weak; they need improvements in many aspects and quality graphics.
7. There are a few minor typographical and grammatical errors.
8. References are not uniform, it must be according to the journal format.

Experimental design

The results are very weak; they need improvements in many aspects and quality graphics on the basis of experimental data. For example, figures 3, 4, and 5 results are very badly visualized.

Validity of the findings

All comments are add above,

Reviewer 2 ·

Basic reporting

In this paper, the authors have conducted a research study on warfarin dosing optimization using different machine learning techniques such as PCA, t-SNE, denoising autoencoder, and Random Forest. They used the MIMIC-III dataset for evaluating the performance of their proposed ML-based techniques. Furthermore, their proposed methodology addresses the dataset's high dimensionality and significant missing data characteristics, enhancing the predictive accuracy of INR levels. This paper uses the root mean square error (RMSE) performance measure to evaluate the performance of the ML-based models utilized. Overall, the quality of the paper is weak. I got the following points for this paper:
1. Replace the paper's title with one that is clearer and more meaningful.

2. The abstract of the paper is too weak. The authors failed in presenting the complete idea and methodology in the abstract of the paper. It needs significant improvements.

3. The introduction section needs a proper revision, as there is no consistency and coherence between sentences and paragraphs.

4. What are the main contributions of your study? Can you please enlist your contributions in bullets form in the introduction of the paper?

5. Usually, there is a paragraph at the end of the introduction section, describing the layout/organization of the paper. Please add paper organization paragraph at the end of the introduction section.

6. The literature review section also needs improvement. The authors have cited very old papers, please cite more recent and relevant papers in the literature review section.

7. Can you put a table in the literature review section, describing the earlier works and the methodology adopted in those studies for sorting out the same problem? Also, can you clearly explain, how your work is better than them!

8. The methodology of the paper needs proper revision. The authors have mainly focused on dimensionality reduction algorithms and ML models. Can you use more advanced dimensionality reduction and ML models for sorting out your proposed problem. These methods are extensively used for sorting out numerous medical problems.

9. What are your scientific contributions in this paper?

10. A proper block diagram is missing in the methodology section of the paper.

11. In the methodology, the authors must focus on their own work and explain it in terms of an algorithm.


12. I would like to recommend the addition of a subsection with the name “performance evaluation measures” in the methodology section of the paper.

13. The results section also needs improvement. Please compare your work with at least 4 recent articles used for solving the same problem.

14. In machine learning and deep learning parameter values has a great influence on the performance of the model. Parameter tuning plays a significant role in a model's performance. The authors should explain what parameters they have used and why they selected the current parameters. The authors should also explain the function of each parameter used in the model.

15. Please improve the conclusion of the paper.

Experimental design

1. In machine learning and deep learning parameter values has a great influence on the performance of the model. Parameter tuning plays a significant role in a model's performance. The authors should explain what parameters they have used and why they selected the current parameters. The authors should also explain the function of each parameter used in the model.

1. The experimental results section needs a lot of improvement. The authors should compare their work with the current SOTA techniques and show that how their work is better than them.

2. How the proposed model is better than the SOTA ML and DL models.

3. In ML and DL it is very crucial to give a clear picture of the experimental design that what parameters, libraries, packages, and tools are used for the implementation of a model. The authors have not provided these information. I suggest adding these information in the results section of the paper.

4. Revise the experiment section by considering these points.

Validity of the findings

Overall, the validity and findings of this paper are not satisfactory.

Additional comments

1. Grammar and formatting of the paper needs improvement.

Reviewer 3 ·

Basic reporting

The literature review should be improved. More up to date works should be added.

Experimental design

More detail should be added on the proposed technique.
The model Figure might be helpful in the proposed technique.
Pre-processing of the data should be in more detail.

Validity of the findings

The results and discussion section needs more elaboration.
More comparison should be added with state of the art works.

Additional comments

The whole manuscript should be checked for grammatical errors.

·

Basic reporting

1. Sections do not follow a coherent and consistent progress. While, for example, “overview” finishes by discussing the families of missing data, the next part (literature review) starts with a discussion on warfarin.
2. Same ideas or talking points, such as the challenges of warfarin dosing, methods of dealing with missing data, etc., repeat at different sections.
3. Some unnecessary details are discussed. For example, the formulation of PCA, t-SNE, etc. are available in the literature and are only needed if the authors are suggesting a modification.
4. Literature review is not up to date. The only 2020+ papers are cited as examples of the application machine learning in different domains. All references related to missing data, warfarin, etc. are relatively old and some go back to as far as 1979! For example, mixed results on smoking and warfarin from 1984 must surely be clarified by now! Or a recommendation on not smoking from 1979 has no value today! Therefore, the latest development in warfarin dosing or use of ML in drug dosing is not covered.
5. None of the tables or charts are referenced in the text. This introduces readability issues. For example, "stages" are not defined properly in the text, but Table 2 provides some definition.

Experimental design

1. Too much details is provided with the methods. Some discussions are useless in the context of the paper. For example, the authors discuss different families of missing data, but simply assume one of the families in their work. Such extra information in not needed as it does not come into play (except a bullet point in the limitations).
2. The paper lacks a number of tables: A list of variables in each dataset (stage) and hyper-parameters as tables (not as lengthy texts).

Validity of the findings

1. This work needs a complete restructuring and re-evaluation. While the title emphasizes "missing data" as goal, and other parts of the text corroborate with it (e.g. line 704, and conclusion section), there are other places that the goal is the use of machine learning in optimizing warfarin dosing (line 54 and 759). This ambiguity is clear in the text and the structure of the paper. In either case, the novelty of this work is not clear to me. Missing data imputation methods are discussed extensively in medical domain. The use of machine learning in warfarin dosing is also prevalent. For example, IWPC did extensive research and used neural nets, SVMs, regression, random forest, etc. in their study and ended up suggesting a non-linear regression model.

2. Some of the conclusions and findings are also lack novelty and originality. For example, authors acknowledge the importance of Factors II, VII, and X, but these are already known. To the extent that the authors in line 693 reject Stage 1 and Stage 2 dataset because of the lack of these features. The lengthy discussion on the outcomes of PCA and t-SNE are also questionable. Both of these methods provide good ways of visualizing data, but their outputs should not be used as definitive evidence to make conclusions. Running t-SNE with different random seeds, different perplexity value, etc. produce totally different charts. And it is a feature of this method to emphasis on local or global structures depending on the perplexity score. So, authors' discussion on this part has no added value.

3. Another questionable section of this draft is the outlier detection. In warfarin dosing, outliers are crucial as sensitive and highly sensitive patients might show very high and dangerous INR values. So, it is critical for a dosing method to address these cases. The use of outlier detection removes such observations. Such elimination of observations improves the prediction model's performance at the expense of neglecting some patients. In fact, INR 1.5 in Stage 3 is a testament of this issue. Health patients have an INR of one and patients on warfarin should be in the range of 2-3. If the average INR is 1.5, it means most patients have not achieved the therapeutic effect of the medication. So, either they are healthy patients or patients in the first days of warfarin administration. In any case, such dataset is not a good candidate for a dose or INR prediction model.

Additional comments

1. Definition and full form of abbreviations should follow the first use of the terms. For example, INR is not defined in the text and its full form is in line 705! PCA and t-SNE are fully mentions in line 180, but used many times before that.
2. INR for highly sensitive patients might go as high as 8 or 9, which can be fatal! Average INR of 54 or 37 are out of this world! There is clearly something wrong, either with the data or the methods.

---

## Round 0.2 · Major Revisions

There are still many comments of the reviewers needs to be addressed. The reviewers have recommend major changes are needed.

Reviewer 1 ·

Basic reporting

The author addresses most of our concerns, so i accept the paper conditionally with minor changes.
1. The author needs to check the paper for minor typos and mistakes thoroughly.
2. Author needs to address the latest related work such as
a. Wang, E., Zhang, M., Yang, B., Yang, Y., & Wu, J. (2024). Large-Scale Spatiotemporal Fracture Data Completion in Sparse CrowdSensing. IEEE Transactions on Mobile Computing, 23(7), 7585-7601. doi: 10.1109/TMC.2023.3339089.
3. All references must be in journal format.

Experimental design

Nil

Validity of the findings

Nil

·

Basic reporting

The revision is quite improved over the initial submission. The text is more coherent and better presented. Literature is relevant and recent. However, there are issues that need to be resolved. For example:
1. End of Section 1.4, “Using the MIMIC-III dataset, the proposed imputation techniques enhance warfarin dosage accuracy, improving patient outcomes Johnson et al. (2016).” I believe that you are citing Johnson on “improving patient outcomes” but combining it with “the proposed imputation techniques” suggests that Johnson has already done these techniques and showed the results. Same happens in 4.1.2, where you write, “The consistent identification of critical coagulation factors, particularly in Stage 3, highlights the model’s capability to provide reliable and precise warfarin dosing recommendations Shah (2020).” Which part of the sentence is backed by Shah?
2. Why do you call your experiments “Stages”? I understand that each is more complicated than the one before, but they are independent experiments, nonetheless. “Stage” implies that a prior stage feeds the latter stage. In Section 4.2, you talk about “Steps” in dimension reduction. It is clear that you progress from Step 1 to Step 2, but “Stages” do not progress. They are different experiments.

Experimental design

Methods are described in details. However, there are issues with the description of the methodology. However, there are issues and concerns:
1. Section 3.4.2, the jump from MICE to GMM was quite abrupt and I couldn’t follow what their relationship is.
2. How do you decide if a variable is CNAR, NAR or MNAR? Was CNAR imputation methods used only for the baseline?
3. When in methodology, do methodology. Do not discuss results on which method performed better.
4. The last sentences of 3.4.1 are repeated.
5. The reader needs a map to know what to expect. Sometimes the path flows naturally, and such a map is not necessary. Other times there are too many stops or too much information that makes the text hard to navigate. The Methodology Section is among the hard-to-navigate ones. You have the map in Section 3.7, but it concludes what came before rather than laying out what comes next! Please add an intro to Section 3 and briefly explain your work so that the reader can put each of the following sections into their proper place.
6. Section 4.1.1 You used Gini Index for “INR prediction”. Why INR, and not dose? In Methodology (line 327), you are predicting dose!

Validity of the findings

1. Despite the initial "Objective" section, the objective seem to drift in the text from INR to dose and vice versa. For example, Section 4.1.3, (line 414) “The model suggests reducing the dose by 10% to prevent excessive INR elevation and bleeding.” Which model? Do you predict INR or dose or percent change in dose? Or, Section 4.1.1 You used Gini Index for “INR prediction”. Why INR, and not dose? In Methodology (line 327), you are predicting dose!
2. Section 4.1 (line 357). Is multicollinearity a concern when you use random forest or other advanced ML methods?
3. What percentage of data is outlier in each stage? From Figure 4, it feels like almost 20-30% are outliers.
4. Neither PCA nor t-SNE are helping. Especially, t-SNE charts simply show how t-SNE works with different perplexity values. In my opinion, it does not have any useful information.
5. [My biggest concern:] Based on your code and your explanation in Section 3.2 (line 205), you applied min-max scaling to all data. Such practice is not acceptable since it is an instance of data leak. You use the information in the training data (minimum and maximum values) in the test data. I understand that min-max scaling is less prone to leak as you can set any arbitrary small (large) number for the minimum (maximum), but it is still not a sound practice to do scaling before splitting.
6. When in methodology, do methodology. Do not discuss results on which method performed better.
7. How do you decide if a variable is CNAR, NAR or MNAR? Was CNAR imputation methods used only for the baseline?

Additional comments

1. The last sentences of 3.4.1 are repeated.
2. Section 3.4.2, the jump from MICE to GMM was quite abrupt and I couldn’t follow what their relationship is.
3. In Table 2 and Figure 2, Inner and Outer Merging are Stages 1 and 2, but in the text (Sections 3.2 and 3.7, for example), it is the opposite.
4. Figure 3, the explanation is incorrect. Blue is positive correlation, red is negative.
5. Figure 4, the axes titles should be “Principle component 1” (and 2), not “Principle component analysis 1”.
6. As a general comment, I assume that you have some patients with complete or near complete information. So, you can manually (different methods) remove data points to create missing data. This way, you have the ground truth and so you can assess the performance of your imputation methods.

---

## Round 0.3 · accepted · Accept

The paper has been improved according to the reviewers comments and suggestions